# OrthoRank: Token Selection via Sink Token Orthogonality for Efficient LLM inference

Seungjun Shin[1]  Jaehoon Oh[1]  Dokwan Oh[1]

## Abstract

Recent studies have revealed the sink token, which receives disproportionately high attention despite its limited semantic role. In this paper, we first explore the relationship between the sink token and other tokens beyond attention, by analyzing their similarity in hidden states. We observe that as layers deepen, the cosine similarity between the normalized hidden states of the sink token and those of other tokens increases, and that the normalized hidden states of the sink token exhibit negligible changes. These imply that other tokens are consistently directed toward the sink token throughout the layers. Next, we propose a dynamic token selection method, called `OrthoRank`, using these findings to select important tokens. Specifically, in a certain layer, we define token importance by the speed at which the token moves toward the sink token. This is converted into orthogonality with the sink token, meaning that tokens that are more orthogonal to the sink token are assigned greater importance. Extensive experiments show that our method results in lower perplexity and higher zero-shot accuracy compared to layer pruning methods at the same sparsity ratio with comparable throughput, while also outperforming on LongBench.

## 1. Introduction

Large language models (LLMs) have shown remarkable performance across various tasks (Thirunavukarasu et al., 2023; Wu et al., 2024; 2023; Labrak et al., 2024; Nam et al., 2024). However, despite this, the computational cost of LLM inference remains a significant challenge, especially for real-time applications.

To address this challenge, many lightweight methods have been proposed for LLMs. Among the various methods, layer pruning is a simple and effective approach to reduce computational costs by removing layers that have less impact on the model. The impact is quantified by either measuring the similarity between the input and output at each layer (Siddiqui et al., 2024; Men et al., 2024), or by evaluating how the removal of a layer effects the final output (Song et al., 2024; Kim et al., 2024). Song et al. (2024) proposed an iterative pruning method based on these metrics, while Kim et al. (2024) introduced a one-shot pruning approach followed by additional tuning using LoRA (Hu et al., 2022).

While effective, layer pruning has inherent limitations. It requires a calibration set to determine which layers can be skipped, applying a fixed pruning decision across all input tokens. This uniformity prevents the method from adapting to token-specific computational needs. At a given layer, some tokens may no longer require further processing, while others still benefit from it. However, the pruning pattern remains static and cannot reflect such variation. As a result, although layer pruning is well suited for environments with limited storage or communication bandwidth, such as on-device inference with small batch sizes, it may fall short in exploiting more fine-grained efficiency gains at the token level.

Motivated by the need for token level processing, early exit (Schuster et al., 2022; Chen et al., 2024b; Bae et al., 2023) and mixture of depth (Raposo et al., 2024) have proposed dynamic computation paths based on token-level characteristics. Early exit determines that a token aligns with the final output, bypassing the remaining layers. Mixture of depth uses routers at each layer to decide whether a token should be computed or skipped. While these methods offer viable solutions, they rely on training additional routers or classifiers, or require the entire model to be trained specifically for early exit. Although these techniques have contributed to LLM acceleration, such as in speculative decoding, their practical use is limited because they require additional training across a wide range of existing models.

Therefore, this paper begins by questioning:

*Can we identify which tokens advantageous to compute at each layer without extra training?*

[1]Samsung Advanced Institute of Technology, Korea. Correspondence to: Seungjun Shin <sj0216.shin@samsung.com>.

*Proceedings of the 42nd International Conference on Machine Learning*, Vancouver, Canada. PMLR 267, 2025. Copyright 2025 by the author(s).

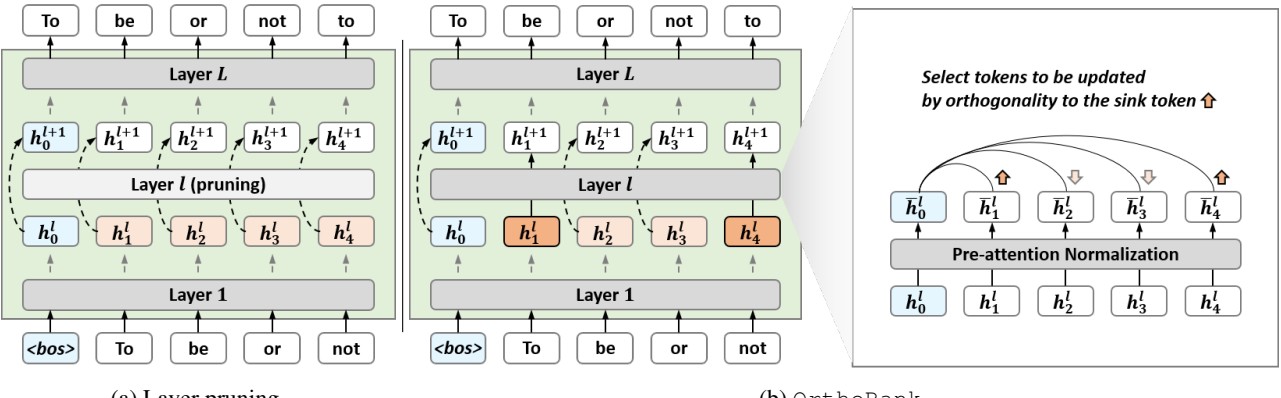

(a) Layer pruning.  (b) `OrthoRank`.

*Figure 1.* Overview of our approach (`OrthoRank`). `OrthoRank` first determines the orthogonality of tokens to the sink token after normalization at each layer. Based on this, the top $K$ tokens are selected for updates, while the remaining tokens bypass the layer except for KV calculations.

To explore this, we analyze the internal workings of LLMs to determine whether each token requires an update within a layer. Our focus is on one of the most distinctive phenomena in LLM behavior: the attention sink (Xiao et al., 2024b), which was first studied by investigating attention distributions and identifying the presence of attention sinks. This phenomenon shows that the initial token in an input sequence receives a disproportionately large share of attention, despite often lacking meaningful semantic value. This occurs because, in autoregressive models, the initial token is visible to nearly all subsequent tokens, leading to 'excessive' attention scores. Since then, this phenomenon has been further explored (Sun et al., 2024; Cancedda, 2024; Gu et al., 2024), calibrated (Yu et al., 2024), and leveraged in various ways (Son et al., 2024; Zhang et al., 2024; Chen et al., 2024a; Tang et al., 2025; Xiao et al., 2024a) to improve LLM efficiency and enhance understanding of their mechanisms. Through further investigation, we observed that the sink tokens and other tokens exhibit a distinctive pattern of cosine similarity (Section 2).

From this, we propose an importance ranking of tokens, `OrthoRank`, which leverages **Ortho**gonality to **Rank** tokens based on their relevance to the sink token. We confirm that selecting tokens with our orthogonal-based importance is effective, as it outperforms the opposite method in language modeling performance (Section 3.1). To apply this across multiple layers in the LLM, we adopt the layer evaluation method from layer pruning. We then replace each layer with a token selection layer and evaluate them to identify the optimal token selection layers (Section 3.2). In Figure 1, we provide an overview of our proposed method, including the selection scheme. The main idea is to calculate each token's orthogonality to the sink token to select tokens. Selected tokens pass through all steps within the layer (e.g., query, key, value, feed forward network, etc.), while unselected tokens only participate in key and value calculations for the selected tokens without updating their own states, similar

to early-exit mechanisms. Many studies (Sun et al., 2024; Son et al., 2024) suggest that in certain models, the attention sink phenomenon occurs not only with any token at the first position but also with specific delimiter tokens (e.g., ".", "\n"). However, for simplicity and consistency, we focus our calculations on the first token. That is, $h_i^l$ represents the input hidden states of the sink token ($i = 0$) and other tokens ($i \geq 1$) at layer $l$.

In summary, the key contributions of our paper are as follows:

- We discover that after the layer where the attention sink occurs, the cosine similarity between the normalized hidden states of the sink token and those of other tokens increases, as the layers deepen. However, the normalized hidden states of the sink token across the layers remains largely unchanged. These mean that other tokens are heading toward the sink token.

- We propose a simple but effective token selection method, `OrthoRank`, based on token-sink orthogonality, prioritizing tokens closer to orthogonality for updates while bypassing others except for KV calculations, without additional modules or training.

- We conduct extensive evaluations demonstrating that our method has better performance compared to the existing layer pruning at the same sparsity with comparable throughput.

## 2. Further Analysis on Attention Sink Beyond Attention

In this section, we revisit the concept of the attention sink and introduce new insights based on further analysis. An attention sink refers to the phenomenon where a particular token receives a disproportionately high amount of attention from other tokens. This phenomenon is always observed after a certain early layer, $l_{sink}$, in the initial token (Xiao et al., 2024b; Sun et al., 2024).

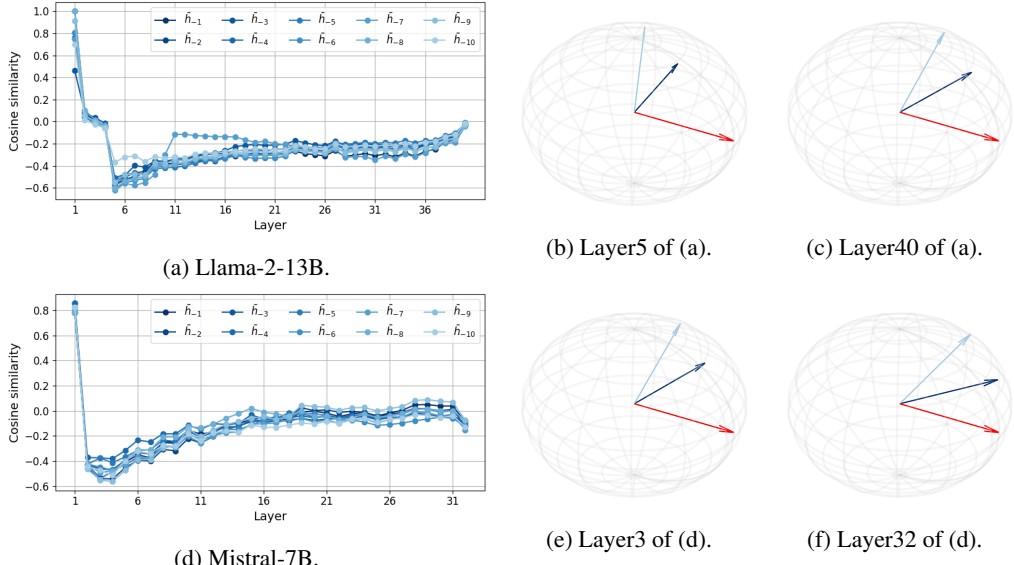

(a) Llama-2-13B.

(b) Layer5 of (a).  (c) Layer40 of (a).

(d) Mistral-7B.

(e) Layer3 of (d).  (f) Layer32 of (d).

*Figure 2.* (a, d) Cosine similarity between the normalized hidden states of the sink token ($\bar{h}_0$) and other tokens of Llama-2-13B and Mistral-7B. $l_{sink}$ is layer 4 and layer 2, respectively. (b-c, e-f) Conceptual representation of the relationship between the sink token (red line) and other tokens (blue lines) at layer right after $l_{sink}$ and the final layer. After the attention sink, as layers progress, the cosine similarity between the sink token and the other tokens increases, indicating that the tokens are gradually aligning more closely with the sink token.

We begin by verifying whether the special relationship between the sink token and the other tokens appears in states other than attention. To the best of our knowledge, we are the first to analyze the behavior of the sink token and other tokens through similarity analysis. In fact, the attention map provides limited information regarding the layers because there is little difference between the layers after layer $l_{sink}$. Therefore, we focus on the hidden states after the pre-attention normalization layer (i.e., the normalized hidden states), because they are the direct inputs for an attention module in each layer.

We investigate the cosine similarity between the normalized hidden states of tokens throughout the layers, using wikitext dataset (Appendix F). Figures 2(a) and 2(d) describe the cosine similarity between the normalized hidden states of the sink token (i.e., $\bar{h}_0$) and those of other tokens (i.e., $\bar{h}_i \ (1 \le i \le 10)$) of Llama-2-13B and Mistral-7B, respectively. For each model, the attention sink occurs at layer 4 and layer 2. It is observed that the cosine similarity between the sink token and other tokens decreases drastically right after layer $l_{sink}$. However, after layer $l_{sink}$, the cosine similarity between the sink token and other tokens tends to increase as the layers progress, although the ranges of cosine similarity vary across models.

These findings are simply illustrated in 3D by Figures 2(b-c) and 2(e-f). We plot each state as a unit vector on the hypersphere to focus on angles at layer right after $l_{sink}$ and the final layer. In the subfigures, the red line represents the normalized hidden states of the sink token (i.e., $\bar{h}_0$),

while the blue lines represent those of other tokens (i.e., $\bar{h}_1$ and $\bar{h}_{10}$). Additionally, the cosine similarity between other tokens except for the sink token does not show any consistent trend, presented in Appendix B. In summary, our findings offer insights that go beyond the information derived from the attention map, revealing that as the layers deepen, the angles between the sink token and the other tokens gradually decrease, after layer $l_{sink}$.

> **Obs. (1).** For layer $l_{sink}$ and the final layer $L$,
> $$cos(\bar{h}_0^{l_{sink}+1}, \bar{h}_i^{l_{sink}+1}) \le cos(\bar{h}_0^L, \bar{h}_i^L), \forall i.$$
> Moreover, when $l_{sink} < l_1 < l_2 \le L$,
> it generally holds $cos(\bar{h}_0^{l_1}, \bar{h}_i^{l_1}) \le cos(\bar{h}_0^{l_2}, \bar{h}_i^{l_2}), \forall i.$

Next, we explore the cosine similarity between the normalized hidden states of the same token across different layers. Through this analysis, we can determine whether the sink token and other tokens are converging towards each other, or if one remains relatively stationary while the other actively moves towards it.

Figures 3(a) and 3(b) illustrate the cosine similarity between the normalized hidden states of the sink token across all layers of Llama-2-13B and Mistral-7B, respectively. The red boundary highlights the layers ranging from $l_{sink}$ to the final layer $L$. For Llama-2-13B, after passing through layer $l_{sink}$, the layers are grouped together, with each group exhibiting a significantly higher degree of similarity (close to 1). Despite this grouping, layers across different groups still maintain

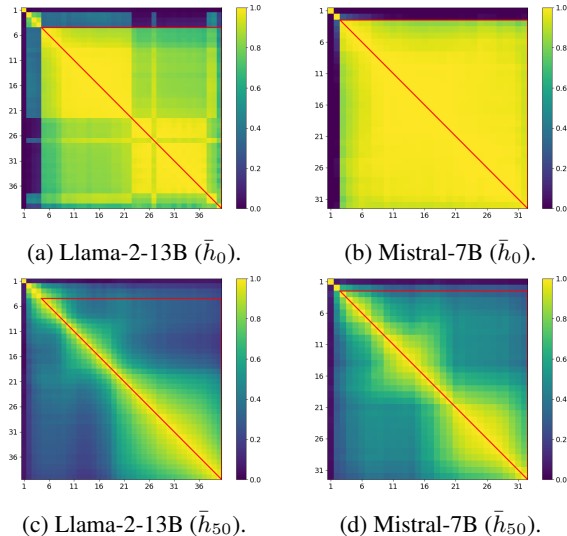

(a) Llama-2-13B ($\bar{h}_0$).      (b) Mistral-7B ($\bar{h}_0$).

(c) Llama-2-13B ($\bar{h}_{50}$).      (d) Mistral-7B ($\bar{h}_{50}$).

*Figure 3.* (a-b) Cosine similarity between the normalized hidden states of the sink token across layers. (c-d) Cosine similarity between the normalized hidden states of another token (postiton 50) across layers. The red boundary represents the layers after layer $l_{sink}$. The sink token shows similar values not only with adjacent layers but also with distant layers, as confirmed through (a) and (b). In contrast, the another token show similarity in adjacent layers, but differences accumulate, leading to dissimilarity in distant layers, as shown in (c) and (d). These results highlight the static nature unique to the sink token, in contrast to other tokens.

a relatively high level of similarity, approaching nearly 0.8. For Mistral-7B, all layers following layer $l_{sink}$ form a single cohesive group, where the similarity between these different layers is consistently close to 1. These results suggest that the sink token experiences almost no change in its trajectory in the normalized hidden states space as it moves through the deeper layers. Therefore, the *fixed* sink token on the hypersphere in Figure 2, which simplifies our Observation (1), is nearly accurate. Furthermore, this observation can be linked to massive activations of the sink token in the hidden states, which appear in a small number of fixed feature dimensions and are delivered to the next layer via the residual connection, keeping high cosine similarity across layers.

Figures 3(c) and 3(d)illustrate the cosine similarity between the normalized hidden states of a token, excluding the sink token, across all layers of Llama-2-13B and Mistral-7B, respectively. As expected, due to the presence of residual skip connections, there is relatively high similarity between adjacent layers, especially along the diagonal. However, as the model processes more layers, differences between the layers begin to accumulate, and the normalized hidden states at the final layer eventually exhibit low cosine similarity compared to the normalized hidden states immediately following layer $l_{sink}$.

> **Obs. (2).** When $l_{sink} < l_1 < l_2 \leq L$,
> $cos(\bar{h}_0^{l_1}, \bar{h}_0^{l_2})$ remains close to 1.
> However, $cos(\bar{h}_i^{l_1}, \bar{h}_i^{l_2})$ decreases
> as the gap between $l_1$ and $l_2$ widens, $\forall i \geq 1$.

From Obs. (1) and Obs. (2), it is concluded that:

*As the layers deepen, other tokens gradually align with the sink token, which remains almost static.*

## 3. OrthoRank: Dynamic token selection

In this section, we extend our observations as criteria for selecting tokens at layer $l$ (Section 3.1). Then, we propose a dynamic token selection algorithm, called OrthoRank (Section 3.2). Our algorithm can be used in conjunction with the layer selection algorithm.

### 3.1. Dynamic token selection criteria

Attention scores are widely used to identify relationships between tokens and are often employed to determine token importance. However, we discover that the relationships between tokens can also be captured through normalized hidden states. Based on this observation, we propose using these states to define token importance.

Our findings suggest that tokens follow a discrete trajectory in which they align with the sink token (i.e., they move in a direction that increases cosine similarity). Building on this, we define the importance of token $i$ in a certain layer after $l_{sink}$ as the speed at which token $i$ can increase its cosine similarity with the sink token[1]:

$$\left\| \frac{\partial}{\partial \bar{h}_i} \cos\left(\bar{h}_0, \bar{h}_i\right) \right\|. \tag{1}$$

Starting from the relation $\bar{h}_0^\top \bar{h}_i = \|\bar{h}_0\| \|\bar{h}_i\| \cos\left(\bar{h}_0, \bar{h}_i\right)$, we compute the gradient of $\cos\left(\bar{h}_0, \bar{h}_i\right)$ with respect to $\bar{h}_i$:

$$\frac{\partial}{\partial \bar{h}_i} \cos\left(\bar{h}_0, \bar{h}_i\right) = \frac{1}{\|\bar{h}_i\|} \left( \frac{\bar{h}_0}{\|\bar{h}_0\|} - \cos\left(\bar{h}_0, \bar{h}_i\right) \frac{\bar{h}_i}{\|\bar{h}_i\|} \right). \tag{2}$$

Assuming that normalized hidden states have approximately equal norms except for sink token, we can simplify the importance of token $i$ based on the cosine similarity. The full derivation is provided in Appendix C:

$$\left\| \frac{\partial}{\partial \bar{h}_i} \cos\left(\bar{h}_0, \bar{h}_i\right) \right\|^2 \propto 1 - \cos^2\left(\bar{h}_0, \bar{h}_i\right). \tag{3}$$

---

[1]For simplicity, $l$ is omitted in this section.

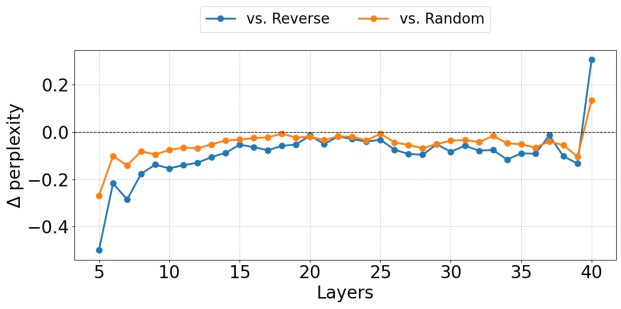

*Figure 4.* Layer-wise performance by token selection criteria. Our method achieves lower perplexity (PPL) across all layers except the final layer compared to both random token selection and the Reverse criteria, which selects tokens in the opposite manner to our approach.

Thus, the importance of token $i$ is directly related to how small $|\cos(\bar{h}_0, \bar{h}_i)|$ is. As $|\cos(\bar{h}_0, \bar{h}_i)|$ decreases, the importance increases because tokens that are **more orthogonal** to the sink token are more likely to be selected, as they have a greater potential to influence the overall cosine similarity.

For implementation convenience, since the norms are approximately equal, we use the absolute value of the inner product $|\bar{h}_0^\top \bar{h}_i|$ as a practical proxy for $|\cos(\bar{h}_0, \bar{h}_i)|$. Therefore, to select the top $k$ important tokens, we rank them based on the smallest $|\bar{h}_0^\top \bar{h}_i|$, which corresponds to **selecting the tokens that are more orthogonal to the sink token**.

$$\text{Select top } k \text{ tokens with smallest } |\bar{h}_0^\top \bar{h}_i|. \quad (4)$$

To validate the effectiveness of our proposed selection criterion, we performed an experiment using the WikiText-2 dataset (Merity et al., 2022). In this experiment, we applied token selection one layer at a time, examining the impact on the model's language modeling performance. For each individual layer, we selected the top 33% of tokens for computation based on our proposed metric, which prioritizes tokens that are more orthogonal to the sink token (i.e., those with the smallest inner product). We then compared the resulting perplexity (ppl) scores to those obtained using an alternative method, where instead of selecting the most orthogonal tokens, we selected the bottom 33%—the tokens with the largest inner product—thereby evaluating the inverse of our approach.

Figure 4 shows the perplexity differences for layers after the attention sink (layer $> 4$), comparing our orthogonal token selection method to both reverse (blue) and random selection (orange). Except for the final layer, our method consistently achieves lower perplexity, indicating superior performance. Layers with higher perplexity occasionally appear, but this varies by model. Model-specific results in Appendix D (Figure 13) further confirm that ours generally outperforms both Random and Reverse (opposite) approaches in most cases.

---

**Algorithm 1** Selecting tokens in an `OrthoRank` layer

**Input:** hidden states $h_o$, where $o$ is an `OrthoRank` layer
**Input:** pruning ratio $p$

```
// Calc. inner product with sink token
```
1   selection_criterion = torch.matmul($h_o$[:,[0],:], $h_o$.transpose(1, 2))
2   selection_criterion = torch.abs(selection_criterion.squeeze(1))

```
// Exception for the sink token
```
3   selection_criterion[:,0] = float(inf)

```
// Sorting for attention module
```
4   lowk_indices = selection_criterion.topk(k=int($p \times h_o$.size(1)),
               largest=False, dim=-1).indices
5   lowk_indices_sorted = torch.sort(lowk_indices).values

```
// Using except key and value states
```
6   selected_hidden_states = torch.gather($h_o$, 1,
    lowk_indices_sorted.unsqueeze(-1).expand(-1, -1, $h_o$.size(-1)))

7   **return** *selected_hidden_states*

---

In summary, our orthogonal token selection criterion leads to better performance across most layers, confirming its effectiveness in reducing computation while maintaining accuracy.

### 3.2. Dynamic token selection with selective layer

In Section 3.1, we demonstrated that selecting tokens closer to orthogonality at each layer improves effectiveness while preserving model performance. However, challenges arise when applying this selection across all layers. First, our selection criteria are less valid before the attention sink occurs. Second, layers near the output are crucial for maintaining model reliability and require computation for most tokens. Additionally, inter-layer dependencies must be considered. Therefore, instead of applying our selection criteria (`OrthoRank`) across all layers, we propose selectively applying it to specific layers. To implement this, we combine our selection criteria with existing layer pruning methods. While traditional layer pruning approaches measure performance by removing layers one by one, we measure performance by applying token selection to layers incrementally. This strategy enables efficient computation across both tokens and layers while preserving model fidelity.

Figure 5 compares layer pruning (Song et al., 2024) and `OrthoRank` with selective layers. In Figure 5(a), Layer pruning is applied to a Llama2-13b model with 40 layers and 20% sparsity, showing the pruned layers. In Figure 5(b), to maintain the same sparsity, 30% of the layers are modified to compute only 33% of the tokens, where the top 33% most orthogonal tokens to the sink token are selected for computation. As shown in Figure 1, only the selected tokens are updated, while the remaining tokens bypass the computation by passing the output of the previous layer

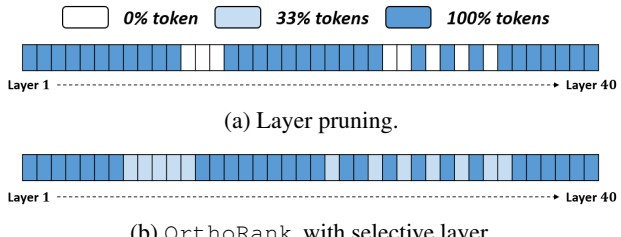

(a) Layer pruning.

(b) `OrthoRank` with selective layer.

*Figure 5.* Comparison under the same sparsity.

through the residual path. Specifically, unselected tokens are excluded from query computation, the query input to the attention mechanism, and the feed-forward network (FFN) computations.

# 4. Experiments

## 4.1. Implementation details

We conducted experiments comparing layer pruning and `OrthoRank` with selective layer approaches. Following the evaluation protocol in (Song et al., 2024), we set target sparsities at 10% and 20%. To ensure the same sparsity ratio across methods, our algorithm applied 15% and 30 % layer selection, with only 33% of tokens computed in the selected layers. We compared our method against two baseline algorithms: the iterative layer pruning method SLEB (Song et al., 2024) and the one-shot pruning method Shortened LLaMA (Kim et al., 2024) without finetuning. Since the one-shot method suffers a significant performance drop at 20% without fine-tuning, we limited its comparison to 10% sparsity. We measured throughput on a single A6000 GPU with a batch size of 32 and prompts of length 2048. The throughput was averaged over 50. To validate the robustness of ours across a wide range of models, we conducted experiments on various models, including Llama2 (7B, 13B, 70B) (Touvron et al., 2023), Llama3 (8B), Llama3.1 (70B) (Dubey et al., 2024), Mistral (7B) (Jiang et al., 2023), Mixtral (8×7B) (Jiang et al., 2024a). Except for the ablation study on token sparsity (token selection ratio) Section 4.7.4), all experiments were conducted with a ratio of 0.333. During generation, we compare each token's inner product with the accumulated context and compute it only if it belongs to the top fixed selection ratio with the smallest values.

## 4.2. Results on Perplexity

Table 1 compares the performance of various models on the language modeling task. Since the layers were pruned (unselected) using the Wikitext-2 dataset, we used the the C4 validation set (Raffel et al., 2020) for the performance comparison. Our proposed method, `OrthoRank`, outperformed other layer pruning approaches in terms of perplexity in most cases.

*Table 1.* Perplexity results on C4 dataset for various models.

| Method | Sparsity | Llama-2 | | | Llama-3 | | Mistral | |
|---|---|---|---|---|---|---|---|---|
| | | 7B | 13B | 70B | 8B | 70B(3.1) | 7B | 8x7B |
| Dense | 0% | 7.26 | 6.73 | 5.71 | 9.45 | 7.11 | 8.38 | 7.41 |
| SLEB | 10% | 8.71 | 7.79 | 6.32 | 12.47 | 8.79 | 9.73 | 8.28 |
| +OrthoRank | 10% | **8.06** | **7.39** | **6.13** | **11.27** | **8.24** | **9.31** | **8.05** |
| SLEB | 20% | 10.90 | 9.42 | 7.31 | 16.49 | 11.18 | 12.39 | 9.50 |
| +OrthoRank | 20% | **10.04** | **8.74** | **7.21** | **14.95** | **10.25** | **11.54** | **9.39** |
| Shortened LLaMA (w/o FT) | 10% | 8.79 | 7.93 | 6.34 | 13.28 | 19.49 | 9.99 | **8.37** |
| +OrthoRank | 10% | **8.04** | **7.60** | **6.29** | **11.22** | **15.58** | **9.43** | 8.47 |

*Table 2.* Mean accuracies (%) on zero-shot tasks for various models evaluated on PIQA, WinoGrande, HellaSwag, ARC-Challenge, and ARC-Easy.

| Method | Sparsity | Llama-2 | | | Llama-3 | | Mistral | |
|---|---|---|---|---|---|---|---|---|
| | | 7B | 13B | 70B | 8B | 70B(3.1) | 7B | 8x7B |
| Dense | 0% | 68.98 | 71.77 | 76.58 | 72.87 | 80.08 | 74.14 | 77.23 |
| SLEB | 10% | 63.13 | 66.74 | 73.13 | 66.94 | 76.22 | 69.04 | 74.60 |
| +OrthoRank | 10% | **65.06** | **69.71** | **74.56** | **69.55** | **76.66** | **69.25** | **75.37** |
| SLEB | 20% | 58.68 | 62.97 | 70.82 | 58.41 | 73.39 | 61.94 | 70.84 |
| +OrthoRank | 20% | **60.35** | **66.99** | **71.25** | **60.84** | **73.96** | **63.88** | **72.52** |
| Shortened LLaMa (w/o FT) | 10% | 62.07 | 69.72 | **74.22** | 69.87 | 64.31 | 66.63 | 71.97 |
| +OrthoRank | 10% | **64.79** | **70.78** | 73.67 | **70.77** | **70.90** | **68.33** | **73.70** |

*Table 3.* Mean accuracies on LongBench for Llama-3-8B.

| Context length for layer selection | Sparsity | Method | Average acc. |
|---|---|---|---|
| | 0% | Dense | 30.93 |
| 2048 | 10% | SLEB | 25.10 |
| | | +OrthoRank | **29.61** |
| | 20% | SLEB | 17.85 |
| | | +OrthoRank | **25.23** |
| 4096 | 10% | SLEB | 25.10 |
| | | +OrthoRank | **29.71** |
| | 20% | SLEB | 17.85 |
| | | +OrthoRank | **24.52** |
| 8192 | 10% | SLEB | **30.42** |
| | | +OrthoRank | 29.77 |
| | 20% | SLEB | 21.53 |
| | | +OrthoRank | **23.89** |

## 4.3. Results on Zero-shot Task

We further evaluated `OrthoRank`'s performance on several zero-shot tasks, including PIQA (Bisk et al., 2020), Wino-Grande (WG) (Sakaguchi et al., 2021), HellaSwag (HS) (Zellers et al., 2019), ARC-easy, and ARC-challenge (Clark et al., 2018), using the LM Evaluation Harness. As shown in Table 2, `OrthoRank` demonstrated better performance compared to layer pruning in most cases.

## 4.4. Results on LongBench

The evaluation results from LongBench (Bai et al., 2024) are shown in Table 3. For the Longbench experiments, we tested calibration lengths of 2048, 4096, and 8192 during the iterative layer selection process. As expected, higher context lengths led to improved performance, and `OrthoRank` consistently outperformed SLEB in most cases.

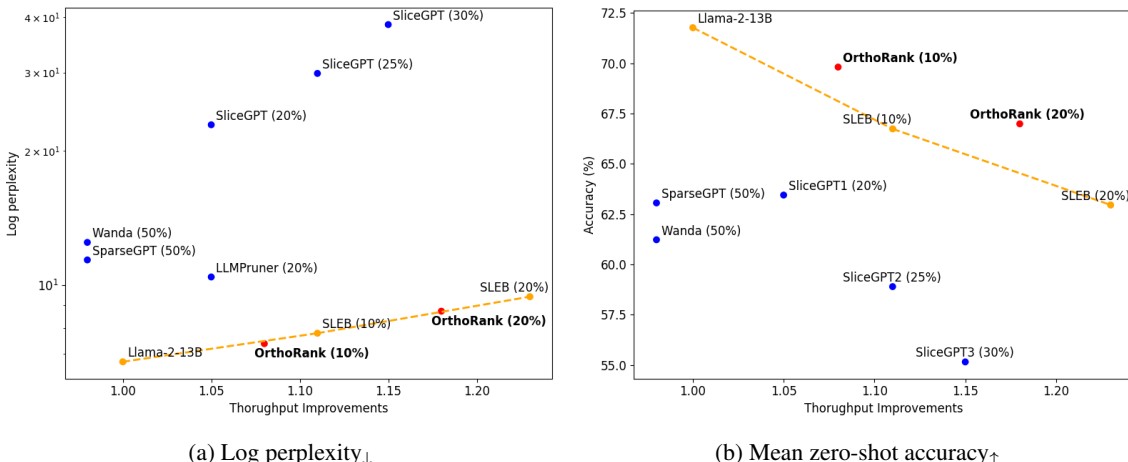

(a) Log perplexity↓            (b) Mean zero-shot accuracy↑

*Figure 6.* Performance vs. Throughput Trade-offs: (a) Log perplexity (↓) and (b) zero-shot accuracy (↑) under throughput improvements.

*Table 4.* TruthfulQA results using MC1 and BLEU scores.

| Method | Llama-2-13B | | Llama-3-8B | | Mistral-7B | | Mixtral-8x7B | |
|---|---|---|---|---|---|---|---|---|
| | mc1↑ | gen↑ | mc1↑ | gen↑ | mc1↑ | gen↑ | mc1↑ | gen↑ |
| SLEB | 21.2 | 20.82 | 19.8 | 4.17 | 21.3 | **21.6** | 24.1 | **27.1** |
| +OrthoRank | **22.3** | **23.6** | **21.6** | **15.28** | **23.6** | 20.17 | **25.2** | 26.85 |

*Table 5.* Ablation study for selection criteria, stage, and KV for unselected token.

| Criteria | Stage | KV | Llama-2-13B | | Llama-3-8B | | Mistral-7B | | Mixtral-8x7B | |
|---|---|---|---|---|---|---|---|---|---|---|
| | | | ppl↓ | acc↑ | ppl↓ | acc↑ | ppl↓ | acc↑ | ppl↓ | acc↑ |
| Random | $\bar{h}_i$ | ✓ | 10.83 | 61.17 | 16.23 | 59.15 | 12.05 | 62.88 | 9.85 | 69.45 |
| Norm ↑ | $\bar{h}_i$ | ✓ | 9.46 | 62.94 | 16.23 | 59.46 | 12.26 | 59.97 | 9.39 | 70.89 |
| Norm ↓ | $\bar{h}_i$ | ✓ | 9.15 | 64.47 | 16.73 | 60.13 | 21.12 | **63.99** | 9.78 | 66.50 |
| Orthogonal ↓ | $\bar{h}_i$ | ✓ | 11.85 | 58.81 | 18.06 | 58.37 | 13.03 | 59.47 | 9.76 | 66.27 |
| **Orthogonal ↑** | $h_i$ | ✓ | 9.64 | 64.30 | 15.72 | 60.82 | **11.53** | 63.87 | 9.55 | 71.49 |
| **Orthogonal ↑** | $\bar{h}_i$ | ✗ | 9.77 | 64.70 | 17.72 | 58.76 | 14.21 | 61.34 | 10.94 | 67.55 |
| **Orthogonal ↑** | $\bar{h}_i$ | ✓ | **8.74** | **66.99** | **14.95** | **60.84** | 11.54 | 63.88 | **9.39** | **72.52** |

## 4.5. Performance vs. Throughput Trade-offs

Figure 6 illustrates the relationship between throughput improvements and performance across different pruning methods using Llama-2-13B. Unlike most pruning algorithms that fail to scale throughput with sparsity, layer pruning methods like SLEB achieve proportional throughput gains. Our proposed algorithm, `OrthoRank` (red), further enhances this efficiency by maintaining a throughput increase nearly proportional to sparsity while achieving performance that is comparable to or even better than SLEB's Pareto optimal curve (orange).

## 4.6. Impact on Generation Quality Metrics

To assess factual quality, we evaluated models on TruthfulQA using two metrics: (1) **MC1**, which selects the highest log-probability choice, and (2) **Generation (BLEU)**, which measures truthfulness and informativeness against ground truth.

As shown in Table 4, `OrthoRank` consistently demonstrates superior MC1 and BLEU scores across most models, suggesting improved factuality and reduced hallucination.

## 4.7. Ablation study

### 4.7.1. TOKEN SELECTION CRITERIA

Row 1, 2, 3, 4 and 7 in Table 5 compare the performance of different token selection strategies: random selection, L2-norm ↑, L2-norm ↓, Orthogonal ↓, and Orthogonal ↑

(Ours). Our method consistently achieves top or comparable results across all models, yielding lower perplexity and higher accuracy than other strategies. While the high-norm strategy slightly outperforms in zero-shot accuracy on Mistral-7B, it suffers from higher perplexity, potentially harming performance elsewhere. In contrast, our method maintains consistently high performance across both metrics, highlighting its robustness. These findings validate the effectiveness of our orthogonality-based criterion introduced in Section 3, enabling more efficient token selection under the same computational budget.

### 4.7.2. SIMILARITY MEASUREMENT STAGE

Table 5 compares the use of hidden states (Row 3) and normalized hidden states (Row 5) for token selection. The results show that using normalized hidden states leads to better performance, with lower perplexity and higher accuracy. As discussed in Section 2, our findings are based on normalized hidden states, making this result consistent with our expectations and further confirming the importance of normalization in improving token selection. However, it performs better than some other components of our approach, suggesting that while the weights within the normalization process do affect the cosine similarity, the hidden state similarity still operates in a somewhat similar manner.

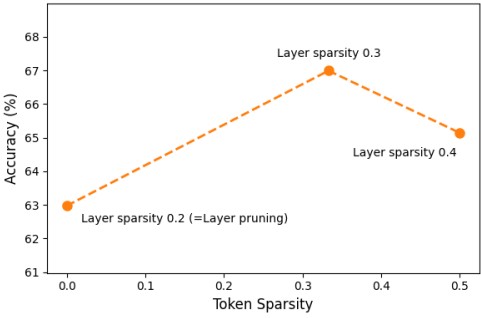

*Figure 7.* Effect of varying token and layer sparsity on Llama-2-13B under a fixed 20% effective sparsity.

### 4.7.3. KV CALCULATION FOR UNSELECTED TOKENS

In Table 5, we compare Row 4, where Key and Value (KV) computations for unselected tokens are skipped, with Row 5, where KV values are computed even for unselected tokens. The results show that calculating KV values for all tokens, regardless of whether they are selected for updates, leads to better performance. This is because our token selection strategy focuses on how quickly a token's state updates, without considering the influence these tokens exert on others through KV interactions. When KV calculations for unselected tokens are skipped, the reduced interaction among tokens significantly degrades overall performance. Therefore, while unselected tokens are not updated, it is essential to compute their KV values to maintain model performance. This approach resembles calculating key-value pairs for tokens that have exited in early exit methods.

### 4.7.4. EFFECT OF TOKEN AND LAYER SPARSITY AT FIXED EFFECTIVE SPARSITY

We conducted experiments using Llama-2-13B by adjusting token and layer sparsity while keeping the overall effective sparsity fixed. As shown in Figure 7, we observe that setting token sparsity in the range of 0 to 0.5 generally yields favorable trade-offs between efficiency and performance. Notably, a token sparsity of 0 effectively corresponds to pure layer pruning, highlighting the importance of balancing sparsity across both dimensions.

### 4.7.5. PERFORMANCE COMPARISON ACROSS SPARSITY LEVELS

Figure 8 compares `OrthoRank`'s performance under varying sparsity levels using log perplexity on the C4 dataset (Figure 8a) and mean accuracy on zero-shot tasks (Figure 8b). `OrthoRank` consistently achieves lower perplexity than SLEB across all sparsity levels except 40%, demonstrating its ability to optimize token selection while preserving language modeling capabilities. In zero-shot tasks,

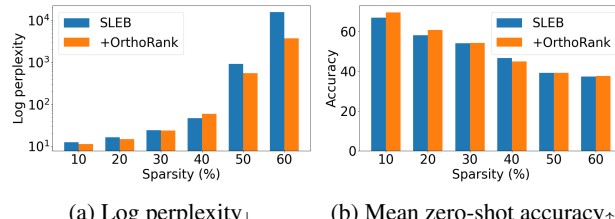

(a) Log perplexity↓      (b) Mean zero-shot accuracy↑

*Figure 8.* Performance comparison across varying sparsity levels with or without `OrthoRank`. (a) Log perplexity (lower is better) on C4 dataset and (b) Mean of zero-shot accuracies (higher is better). `OrthoRank` demonstrates superior performance across all sparsity levels except at 0.4.

`OrthoRank` also outperforms other methods across most sparsity levels, except at 40%, where performance parity is observed. These results highlight the effectiveness of token orthogonality-based selection in balancing computational efficiency and model accuracy across diverse tasks.

*Table 6.* Comparison of token selection criteria at 20% sparsity across models. `OrthoRank` selects tokens based on hidden state orthogonality, while the attention-based baseline relies on attention scores, incurring higher computational overhead.

| Criteria | Throughput improvement | Llama-2-13B ppl↓ | acc↑ | Llama-3-8B ppl↓ | acc↑ | Mistral-7B ppl↓ | acc↑ | Mixtral-8x7B ppl↓ | acc↑ |
|---|---|---|---|---|---|---|---|---|---|
| **Orthogonal** ↑ | **1.18×** | **8.74** | **66.99** | 14.95 | **60.84** | **11.54** | **63.88** | 9.39 | **72.52** |
| Attention ↑ | 0.71× | 8.90 | 63.33 | **14.85** | 57.76 | 11.60 | 56.81 | **9.01** | 66.73 |

### 4.7.6. COMPARISON WITH ATTENTION-BASED SELECTION

To further validate our orthogonality-based token selection criterion, we additionally compare `OrthoRank` with an attention-based baseline (Table 6). Unlike hidden-state-level criteria (e.g., random, L2-norm, orthogonality), attention-based selection explicitly computes attention scores for all tokens, thereby introducing substantial overhead. Additionally, attention-based methods treat tokens primarily as sources of information, prioritizing tokens highly attended by others to preserve the attention matrix structure. While high attention scores indicate a token's strong influence, they offer limited insight into the necessity of updating that token itself.

As shown in Table 6, attention-based selection consistently results in lower zero-shot accuracy across models, despite occasionally achieving competitive perplexity. Moreover, the computational overhead significantly reduces throughput (0.71×) and limits compatibility with optimized fused-kernel implementations (e.g., FlashAttention, SDPA). In contrast, `OrthoRank` achieves consistently competitive performance in both perplexity and accuracy, providing meaningful throughput gains (1.18×) without such computational drawbacks.

# 5. Related work

**Layer Pruning.**   Layer pruning has been a prominent approach for reducing the computational complexity of large language models (LLMs), particularly in transformer architectures (Siddiqui et al., 2024; Men et al., 2024). Recent approaches such as SLEB (Song et al., 2024) and Shortened LLaMA (Kim et al., 2024) aim to remove entire layers that are deemed less critical for downstream tasks. These methods often rely on performance metrics or sensitivity analysis to determine which layers contribute less to overall model accuracy and can be pruned without significant loss of performance.

While layer pruning is an effective way to reduce model depth and memory usage, especially in storage-constrained or low-batch settings such as on-device deployment, it can cause sharp performance drops when critical layers are removed. Since it operates at the granularity of entire layers, it does not reflect token-level variation. In our approach, we combine layer pruning and token selection by applying token-wise sparsity within a subset of layers selected via pruning.

**Token Pruning.**   Token pruning methods have been widely explored as a way to reduce the number of tokens processed across layers, thus decreasing computational load. Recent advanced techniques such as dynamic token selection (Lou et al., 2024) and early exit mechanisms (Chen et al., 2024b; Del Corro et al., 2023; Elhoushi et al., 2024; Bae et al., 2023) progressively drop tokens deemed uninformative as they pass through layers. These methods rely on criteria such as attention scores or token contribution measures to decide which tokens to prune.

However, one potential downside of token pruning is the loss of potentially relevant information as tokens are eliminated layer by layer, especially in deeper models where remaining tokens may not fully capture the complexity of the input sequence. Our approach differs significantly in that we do not progressively drop tokens across layers. Instead, we selectively compute a subset of tokens at specific layers based on their orthogonality to the sink token. This ensures that we preserve the flexibility to compute tokens based on their relevance without completely discarding them, thus mitigating the risk of information loss while still reducing computational costs.

Concurrent work such as D-LLM (Jiang et al., 2024b) also departs from traditional token pruning by dynamically skipping computation without fully discarding tokens, similar in spirit to our approach. Similarly, Mixture-of-Depths (Raposo et al., 2024) learns to route tokens across layers using trainable routers under a fixed compute budget. D-LLM also introduces decision modules at each layer that learn, via Gumbel-Softmax, whether to execute or skip computation for each token, typically fine-tuned using LoRA. However,

both approaches require training of routing components, and Mixture-of-Depths must train the entire model from scratch, limiting compatibility with pretrained language models. In contrast, OrthoRank requires no router training or fine-tuning, and can be directly applied to existing models. Its selection criterion, based on cosine orthogonality to the sink token, provides a simple and interpretable mechanism for controlling computation at the token level.

**Attention Sink.**   The concept of the attention sink, where certain tokens receive disproportionately high attention across layers, has gained attention in recent studies (Sun et al., 2024; Cancedda, 2024; Gu et al., 2024; Yu et al., 2024; Son et al., 2024; Zhang et al., 2024; Chen et al., 2024a; Tang et al., 2025; Xiao et al., 2024a). Xiao et al. (2024b) first introduced the term "attention sink" to describe how the initial token in a sequence tends to dominate attention scores in autoregressive models. This is attributed to its visibility to all subsequent tokens, causing it to act as a "sink" for attention. Sun et al. (2024); Gu et al. (2024) conducted an in-depth investigation into the attention sink phenomenon, revealing that this behavior emerges as a result of massive activation and layer normalization. Furthermore, they discovered that the severity of this phenomenon can be alleviated through the application of KV bias.

Building on these observations, our work explores token-sink orthogonality and uses this metric to inform token selection. By selecting tokens that are more orthogonal to the sink token, we prioritize tokens with greater potential to contribute to meaningful computations, leveraging the inherent token dynamics to optimize inference efficiency.

# 6. Conclusion

In this paper, we introduced `OrthoRank`, a dynamic token selection strategy based on the orthogonality between tokens and the sink token. Our approach was motivated by the observation that as layers deepen, tokens increasingly align with the sink token in the normalized hidden state space. By analyzing token-sink similarity, we found that tokens more orthogonal to the sink token play a greater role in computation. Leveraging this insight, we developed a token selection mechanism that prioritizes such tokens at specific layers, leading to more efficient computation. By applying this token selection approach to selective layers, we achieved superior performance compared to traditional layer pruning methods at the same sparsity level with comparable throughput. Extensive experiments demonstrated significant improvements, and ablation studies confirmed that our selection scheme is optimized both theoretically and empirically. Furthermore, our findings on token-sink similarity offer valuable insights for future research in Efficient LLM inference and Interpretable LLMs, providing a foundation for further optimizing and understanding large language models.

## Impact Statement

This paper presents work whose goal is to advance the field of Machine Learning. There are many potential societal consequences of our work, none which we feel must be specifically highlighted here.

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

# A. Cosine similarity changes of the same token across layers

We investigate cosine similarity changes of the same token across layers using various models, including Llama-2-7B, 13B, and 70B; Meta-Llama-3-8B; and Llama-3.1-70B. as shown in Figure 10.

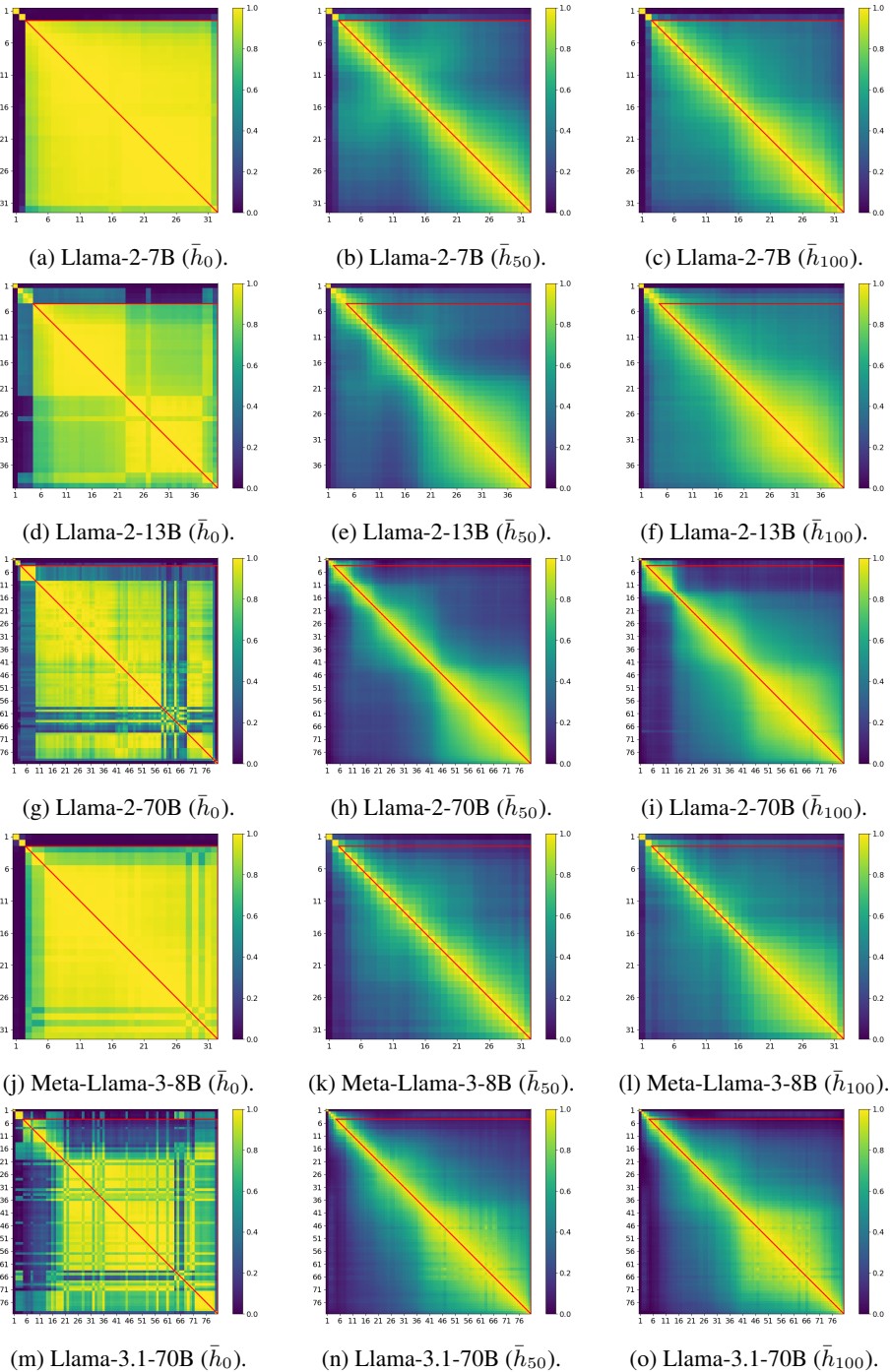

*Figure 9.* Cosine similarity across layers for different models and token positions. Each row represents a token position: 0 (sink), 50, and 100. The red boundary represents the layers after layer $l_{sink}$. The sink token shows similar values not only with adjacent layers but also with distant layers. In contrast, other tokens show similarity in adjacent layers, but differences accumulate, leading to dissimilarity in distant layers These results highlight the static nature unique to the sink token, in contrast to other tokens.

# B. Cosine similarity changes between tokens across layers

We investigate cosine similarity changes between tokens across layers using various models, including Llama-2-7B, 13B, and 70B; Meta-Llama-3-8B; and Llama-3.1-70B. The results are shown in Figure 10, which presents the similarity between position 0 (sink) and positions 91–100, as well as between position 50 and positions 91–100. Figure 11 shows the similarity between position 0 (sink) and positions {1,10,100,1000}, as well as between position 500 and positions {1,10,100,1000}.

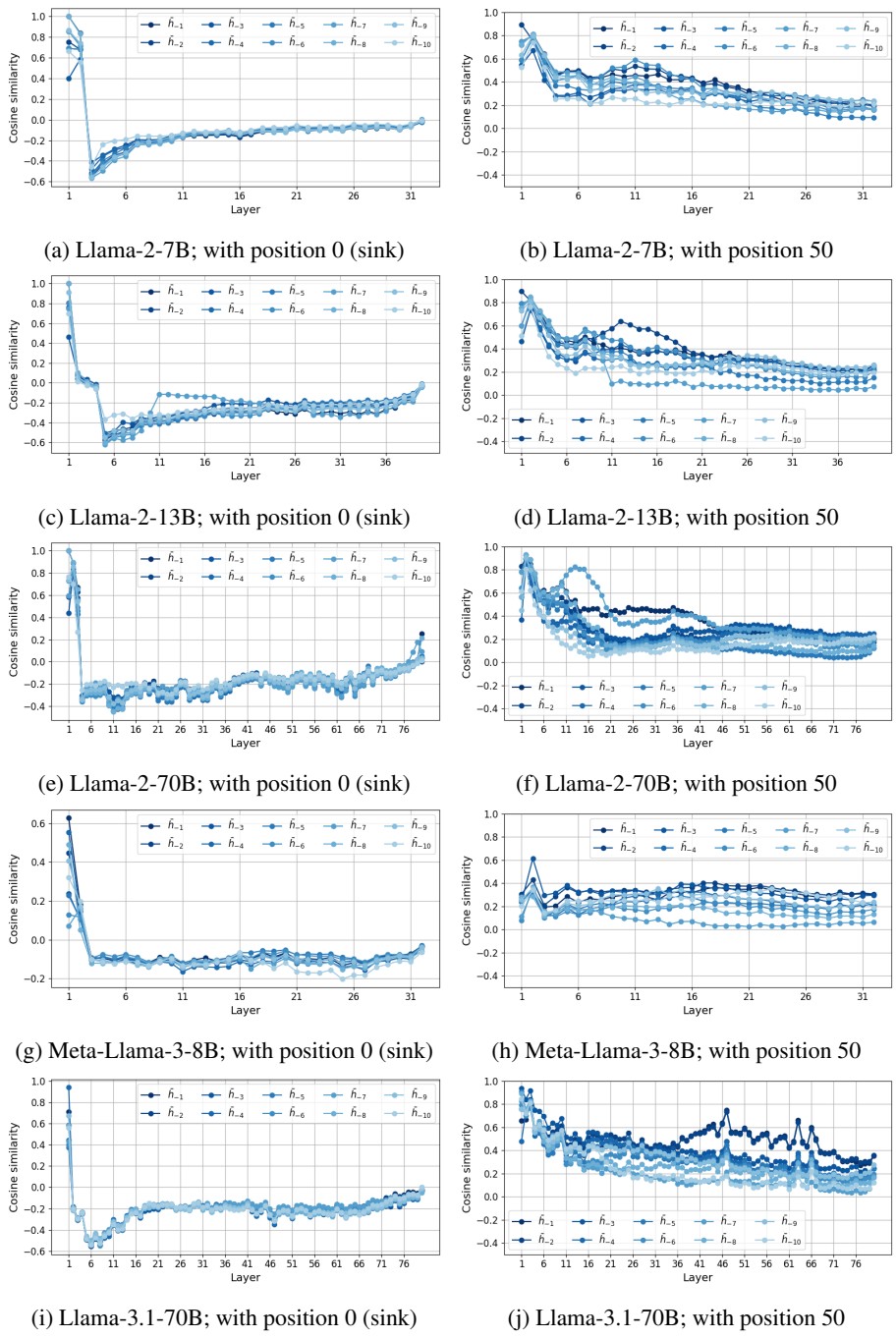

(a) Llama-2-7B; with position 0 (sink)  (b) Llama-2-7B; with position 50

(c) Llama-2-13B; with position 0 (sink)  (d) Llama-2-13B; with position 50

(e) Llama-2-70B; with position 0 (sink)  (f) Llama-2-70B; with position 50

(g) Meta-Llama-3-8B; with position 0 (sink)  (h) Meta-Llama-3-8B; with position 50

(i) Llama-3.1-70B; with position 0 (sink)  (j) Llama-3.1-70B; with position 50

*Figure 10.* Cosine similarity changes between tokens across layers. Each column corresponds to a different model: Llama-2-7B, 13B, 70B; Meta-Llama-3-8B; and Llama-3.1-70B. Left: Cosine similarity between tokens at position 0 and 91–100. Right: Cosine similarity between tokens at position 50 and 91–100.

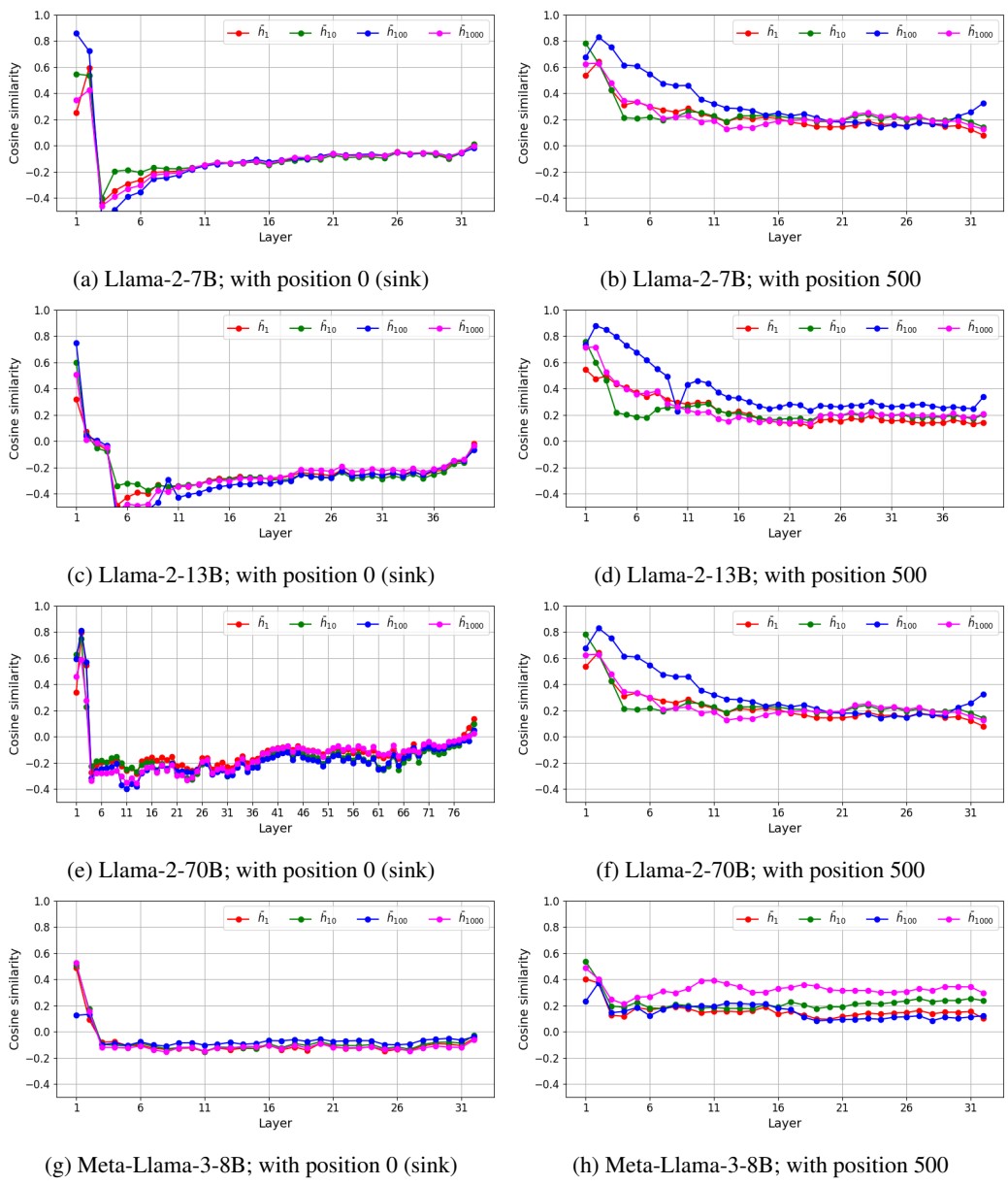

(a) Llama-2-7B; with position 0 (sink)

(b) Llama-2-7B; with position 500

(c) Llama-2-13B; with position 0 (sink)

(d) Llama-2-13B; with position 500

(e) Llama-2-70B; with position 0 (sink)

(f) Llama-2-70B; with position 500

(g) Meta-Llama-3-8B; with position 0 (sink)

(h) Meta-Llama-3-8B; with position 500

*Figure 11.* Cosine similarity changes between diverse tokens across layers. Each column corresponds to a different model: Llama-2-7B, 13B, 70B; and Meta-Llama-3-8B. Left: Cosine similarity between tokens at position 0 and $\{1, 10, 100, 1000\}$. Right: Cosine similarity between tokens at position at position 50 and $\{1, 10, 100, 1000\}$. Results highlight that tokens, even when their positions are far from each other, show increasing cosine similarity with the sink token (position 0) across layers (left), but show no consistent trend with position 50 (right).

## C. Supplementary Derivation for Section 3

We begin with the gradient of the cosine similarity between the sink token's hidden state $\bar{h}_0$ and the hidden state of token $i$, $\bar{h}_i$

$$\frac{\partial}{\partial \bar{h}_i} \cos\left(\bar{h}_0, \bar{h}_i\right) = \frac{1}{\|\bar{h}_i\|} \left( \frac{\bar{h}_0}{\|\bar{h}_0\|} - \cos\left(\bar{h}_0, \bar{h}_i\right) \frac{\bar{h}_i}{\|\bar{h}_i\|} \right). \tag{5}$$

We are interested in the magnitude of this gradient, specifically its squared norm:

$$\left\| \frac{\partial}{\partial \bar{h}_i} \cos\left(\bar{h}_0, \bar{h}_i\right) \right\|^2 = \frac{1}{\|\bar{h}_i\|^2} \left\| \frac{\bar{h}_0}{\|\bar{h}_0\|} - \cos\left(\bar{h}_0, \bar{h}_i\right) \frac{\bar{h}_i}{\|\bar{h}_i\|} \right\|^2 \tag{6}$$

As observed in Figure 12, the norms of the hidden states $\|\bar{h}_i\|$ (excluding $\|\bar{h}_0\|$) follow the behavior described in Section C.1. Therefore, we assume:

$$\|\bar{h}_i\| \approx c \tag{7}$$

Substituting $\|\bar{h}_i\| \approx c$ into the gradient norm squared:

$$\left\| \frac{\partial}{\partial \bar{h}_i} \cos(\bar{h}_0, \bar{h}_i) \right\|^2 \approx \frac{1}{c^2} \left\| \frac{\bar{h}_0}{\|\bar{h}_0\|} - \cos(\bar{h}_0, \bar{h}_i) \frac{\bar{h}_i}{c} \right\|^2 \tag{8}$$

To simplify the expression, we define unit vectors $\mathbf{u}, \mathbf{v}$ as follows:

$$\mathbf{u} = \frac{\bar{h}_0}{\|\bar{h}_0\|}, \quad \mathbf{v} = \frac{\bar{h}_i}{c} \tag{9}$$

Substituting $\mathbf{u}$ and $\mathbf{v}$ into the expression:

$$\left\| \frac{\partial}{\partial \bar{h}_i} \cos(\bar{h}_0, \bar{h}_i) \right\|^2 \approx \frac{1}{c^2} \|\mathbf{u} - \cos(\mathbf{u}, \mathbf{v})\mathbf{v}\|^2 \tag{10}$$

We compute the squared norm:

$$\|\mathbf{u} - \cos(\mathbf{u}, \mathbf{v})\mathbf{v}\|^2 = \|\mathbf{u}\|^2 - 2\cos(\mathbf{u}, \mathbf{v})\mathbf{u}^\top \mathbf{v} + \cos^2(\mathbf{u}, \mathbf{v})\|\mathbf{v}\|^2$$
$$\approx 1 - 2\cos^2(\mathbf{u}, \mathbf{v}) + \cos^2(\mathbf{u}, \mathbf{v})$$
$$= 1 - \cos^2(\mathbf{u}, \mathbf{v})$$

Substituting back:

$$\left\| \frac{\partial}{\partial \bar{h}_i} \cos(\bar{h}_0, \bar{h}_i) \right\|^2 \approx \frac{1}{c^2} \left( 1 - \cos^2(\bar{h}_0, \bar{h}_i) \right) \tag{11}$$

Since $\frac{1}{c^2}$ is a constant, the gradient norm squared is proportional to $1 - \cos^2(\bar{h}_0, \bar{h}_i)$:

$$\left\| \frac{\partial}{\partial \bar{h}_i} \cos(\bar{h}_0, \bar{h}_i) \right\|^2 \propto 1 - \cos^2(\bar{h}_0, \bar{h}_i) \tag{12}$$

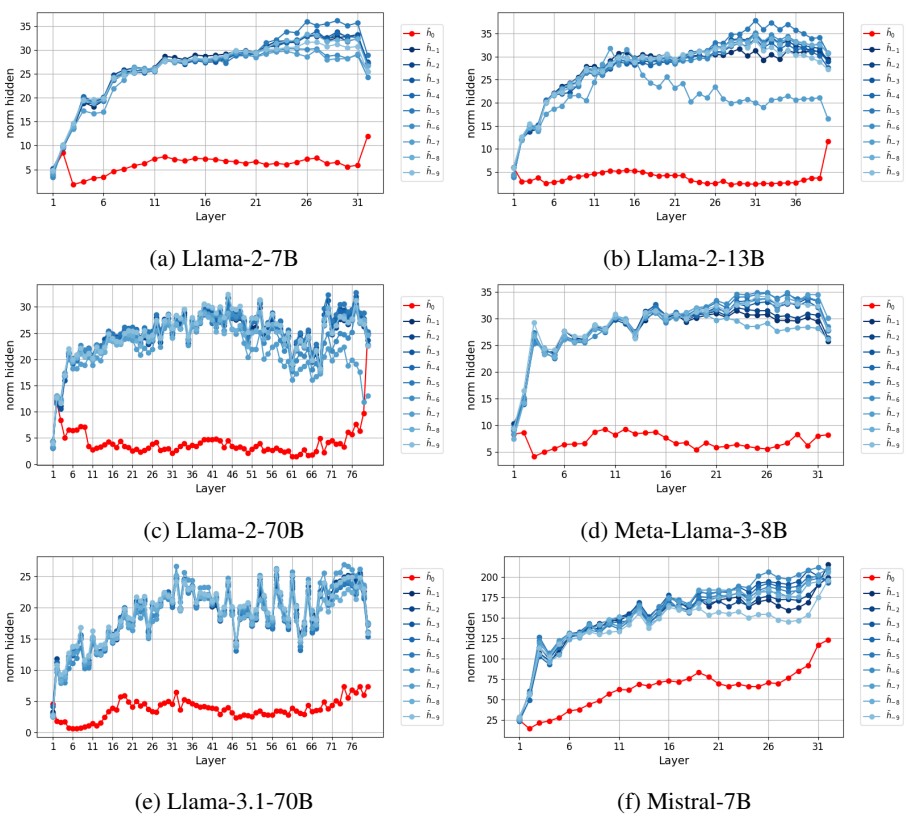

*Figure 12.* Norms of the normalized hidden states. This figure plots the norms of the normalized hidden states $\|\bar{h}_i\|$ for various token positions in the sequence. The red line represents the norm of the hidden state at position 0 (the sink token), while the blue lines correspond to the norms of the hidden states at positions 91 to 100. The plot illustrates that the norms of the hidden states (excluding the sink token) are approximately equal.

### C.1. Norm Behavior under RMS Scaling

`OrthoRank` operates on scaled, normalized hidden states derived via RMSNorm. RMSNorm normalizes each token vector to unit RMS norm and then applies a learned element-wise scale:

$$\mathbf{h}_i^{\text{scaled}} = \mathbf{g} \odot \left( \frac{\mathbf{h}_i}{\text{RMS}(\mathbf{h}_i)} \right) = \mathbf{g} \odot \mathbf{h}_i^{\text{norm}} \tag{13}$$

The scaling vector $\mathbf{g}$ has low empirical variance (approximately 0.001), introducing only minimal distortion in token-wise norms. For most tokens (non-sink tokens), the cosine similarity between $\mathbf{h}_i^{\text{scaled}}$ and $\mathbf{h}_i^{\text{norm}}$ exceeds 0.95:

$$\cos(\theta) \approx 1 \quad \Rightarrow \quad \mathbf{h}_i^{\text{scaled}} \approx \alpha \mathbf{h}_i^{\text{norm}} \tag{14}$$

This indicates that scaling behaves approximately as scalar multiplication. Since scalar multiplication preserves norm ratios, the scaled hidden states maintain near-uniform norms across non-sink tokens:

$$\|\mathbf{h}_i^{\text{scaled}}\| \approx \alpha \cdot c \approx \|\mathbf{h}_j^{\text{scaled}}\| \quad \text{for all } i, j \text{ in non-sink tokens} \tag{15}$$

A small subset of tokens (sink tokens) shows lower similarity (e.g., ~0.6) and significant deviation in norm, often accompanied by concentrated activation in a few dimensions. Empirical observations supporting these behaviors are shown in Figure 12.

# D. Layer-wise Performance by Token Selection Criteria

Figure 13 shows the perplexity differences for layer-wise manner, comparing our orthogonal token selection method to both reverse (green) and random selection (blue). OrthoRank demonstrated superior performance compared to both Random and Reverse (opposite) approaches in most models and across most layers. Rare layers where OrthoRank performs worse are automatically filtered out during the selection process, so this does not pose a significant issue.

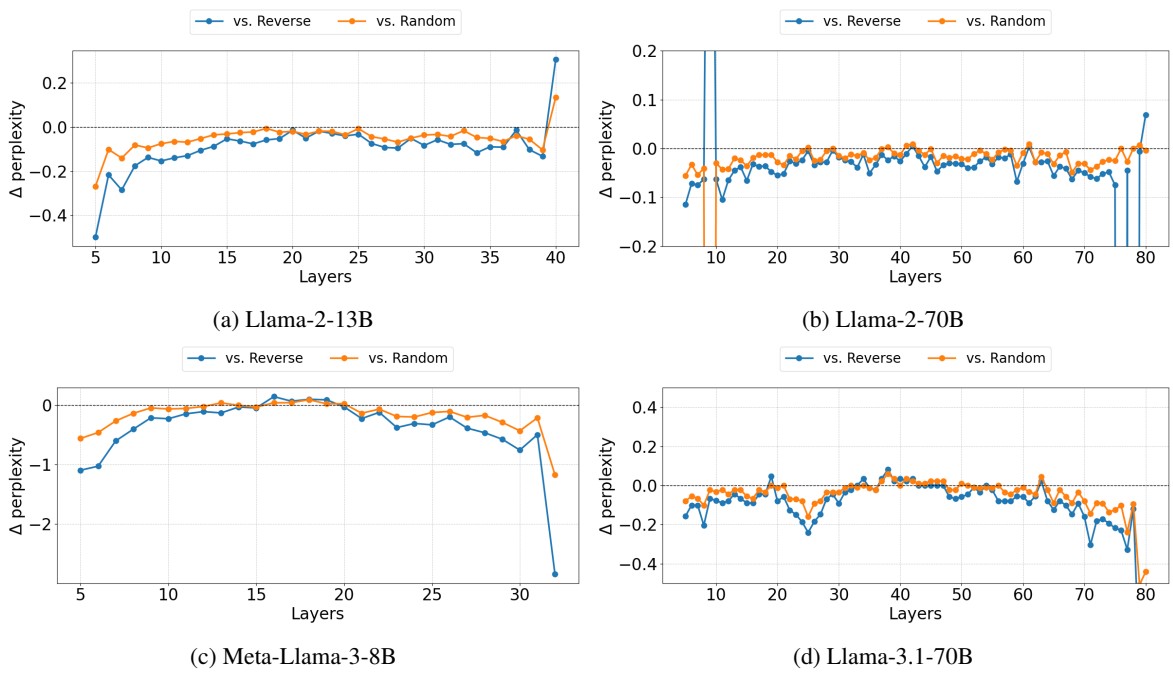

(a) Llama-2-13B      (b) Llama-2-70B

(c) Meta-Llama-3-8B      (d) Llama-3.1-70B

*Figure 13.* Layer-wise Performance by Token Selection Criteria

# E. Results for each task

In this section, we present the detailed results for each task corresponding to Table 2, and Table 3 in the main paper.

*Table 7.* Zero-shot results (%) for various tasks and models with SLEB and OrthoRank by sparsity.

| Model | Sparsity | Method | PIQA | WG | HS | ARC-C | ARC-E |
|---|---|---|---|---|---|---|---|
| Llama-2-7B | 10% | SLEB | 77.15 | 63.38 | 70.69 | 38.57 | 65.87 |
| | | +OrthoRank | **77.97** | **65.82** | **72.22** | **40.96** | **69.32** |
| | 20% | SLEB | 74.54 | **59.51** | 64.71 | 35.41 | 59.22 |
| | | +OrthoRank | **76.33** | 56.51 | **65.68** | **37.12** | **65.95** |
| Llama-2-13B | 10% | SLEB | 79.11 | 66.85 | 74.33 | 41.55 | 71.89 |
| | | +OrthoRank | **79.82** | **69.93** | **75.96** | **46.59** | **76.26** |
| | 20% | SLEB | 76.67 | **65.11** | 70.52 | 38.23 | 64.35 |
| | | +OrthoRank | **79.16** | 64.17 | **73.00** | **44.45** | **74.16** |
| Llama-2-70B | 10% | SLEB | **81.50** | 75.06 | 80.00 | 52.30 | 76.77 |
| | | +OrthoRank | 81.34 | **75.69** | **81.46** | **55.72** | **78.62** |
| | 20% | SLEB | 80.14 | **73.09** | 77.20 | 48.29 | 75.38 |
| | | +OrthoRank | **80.47** | 73.01 | **78.42** | **50.00** | **76.68** |
| Meta-Llama-3-8B | 10% | SLEB | 78.02 | 67.88 | 71.79 | 44.37 | 72.56 |
| | | +OrthoRank | **79.49** | **69.30** | **71.88** | **47.70** | **74.62** |
| | 20% | SLEB | 75.19 | 56.59 | 63.35 | 35.49 | 61.11 |
| | | +OrthoRank | **75.52** | **58.01** | **65.63** | **38.40** | **66.67** |
| Llama-3.1-70B | 10% | SLEB | 83.35 | **76.09** | 81.78 | 58.19 | 81.70 |
| | | +OrthoRank | **83.84** | 71.27 | **82.84** | **60.67** | **84.68** |
| | 20% | SLEB | 81.39 | **74.03** | 78.00 | 53.67 | 79.12 |
| | | +OrthoRank | **82.10** | 69.77 | **79.87** | **56.48** | **81.56** |
| Mistral-7B | 10% | SLEB | 79.22 | 68.59 | 76.60 | 45.65 | 74.03 |
| | | +OrthoRank | **80.96** | **70.72** | **77.49** | **46.25** | **74.71** |
| | 20% | SLEB | 76.88 | 62.75 | 66.71 | 37.46 | 64.14 |
| | | +OrthoRank | **77.48** | **65.27** | **70.91** | **40.02** | **65.74** |
| Mixtral-8x7B | 10% | SLEB | 82.81 | 72.70 | 81.35 | 54.95 | 80.26 |
| | | +OrthoRank | **82.97** | **74.51** | **82.16** | **58.79** | **81.57** |
| | 20% | SLEB | 80.47 | 71.20 | 77.33 | 48.89 | 76.35 |
| | | +OrthoRank | **81.94** | **72.85** | **79.24** | **52.05** | **78.16** |

*Table 8.* Longbench performance comparison across tasks with varying context lengths of calibration and sparsity.

| Context Length | Sparsity | Method | Datasets | | | | | | | | | | | | | | | |
|---|---|---|---|---|---|---|---|---|---|---|---|---|---|---|---|---|---|---|
| | | | NrtvQA | Qasper | MF-en | HotpotQA | 2WikiMQA | Musique | GovReport | QMSum | MultiNews | TREC | TriviaQA | SAMSum | PCount | PRe | Lcc | RB-P |
| | Dense | | 17.13 | 14.08 | 24.93 | 10.09 | 12.4 | 6.77 | 30.57 | 23.72 | 1.6 | 71.5 | 89.9 | 45.6 | 1.16 | 9.5 | 70.47 | 65.53 |
| 2048 | 10% | SLEB | 5.87 | 6.26 | 17.93 | 7.84 | 8.63 | 4.73 | 24.83 | 19.72 | **19.58** | 47.5 | 82.94 | 39.11 | **3.14** | 3.23 | 58.03 | 52.31 |
| | | +OrthoRank | **17.35** | **11.45** | **23.84** | **9.5** | **12.67** | **7.03** | **28.73** | **22.62** | 1.27 | **65.5** | **90.82** | **43.37** | 0.29 | **10.92** | **66.26** | **63.77** |
| | 20% | SLEB | 1.91 | 5.01 | 14.27 | 5.23 | 6.87 | 3.57 | 14.14 | 16.58 | **7.54** | 48 | 46.98 | 24.42 | **1.34** | **5.27** | 42.67 | 41.93 |
| | | +OrthoRank | **11.84** | **9.05** | **20.59** | **8.34** | **10.54** | **4.88** | **19.58** | **22.75** | 1.69 | **48.5** | **86.73** | **40.42** | 1.31 | 4.72 | **55.65** | **57.15** |
| 4096 | 10% | SLEB | 5.87 | 6.26 | 17.93 | 7.84 | 8.63 | 4.73 | 24.83 | 19.72 | **19.58** | 47.5 | 82.94 | 39.11 | **3.14** | 3.23 | 58.03 | 52.31 |
| | | +OrthoRank | **17.35** | **11.45** | **23.84** | **9.5** | **12.67** | **7.03** | **28.73** | **22.62** | 1.27 | **65.5** | **90.82** | **43.37** | 0.29 | **10.92** | **66.26** | **63.77** |
| | 20% | SLEB | 1.91 | 5.01 | 14.27 | 5.23 | 6.87 | 3.57 | 14.14 | 16.58 | **7.54** | 48 | 46.98 | 24.42 | **1.34** | **5.27** | 42.67 | 41.93 |
| | | +OrthoRank | **10.86** | **8.45** | **19.07** | **8.64** | **10.25** | **4.76** | **19.08** | **22.87** | 1.3 | 40.5 | **86.23** | **40.9** | 1.32 | 4.9 | **56.31** | **56.97** |
| 8192 | 10% | SLEB | 16.98 | **13.71** | **24.44** | 9.16 | **12.87** | 6.25 | **29.59** | 20.54 | **15.19** | 70.5 | 87.17 | 42.52 | **2** | 7 | 66.11 | 62.65 |
| | | +OrthoRank | **17.25** | 11.89 | 22.64 | **9.29** | 12.45 | **7.23** | 28.08 | **23.15** | 1.54 | 68 | **90.98** | **42.99** | 0.42 | **8.88** | **67.17** | **64.29** |
| | 20% | SLEB | 3.68 | 7.6 | **19.96** | 5 | 8.6 | 3.42 | 19.27 | 17.25 | **8.24** | 59 | 75.68 | 33.16 | **3.14** | 3.6 | 35.34 | 41.49 |
| | | +OrthoRank | **12.54** | **7.91** | 19.75 | **8.76** | **10.57** | **5.01** | **19.45** | **21.45** | 1.22 | 30.5 | **87.9** | **39.84** | 1.1 | **3.9** | **56.28** | **56.09** |

# F. Text Inputs Used for Figures 2, 3, 9, and 10.

This section provides the input text used to generate the plots in Figures 2, 3, 9, and 10. Due to memory constraints associated with storing token hidden states for visualization, we limited the context length to 101 tokens. The following text is extracted as the first 101 tokens from the test split of WikiText-2-raw-v1. For clearer visualization, we excluded \n , which is known to cause additional attention sink (Sun et al., 2024), from the experiments.

---

= Robert Boulter = Robert Boulter is an English film , television and theatre actor . He had a guest @-@ starring role on the television series The Bill in 2000 . This was followed by a starring role in the play Herons written by Simon Stephens , which was performed in 2001 at the Royal Court Theatre . He had a guest role in the television series Judge John Deed in 2002 . In 2004 Boulter landed a role as " Craig " in the episode " Teddy 's Story " of the television series The Long Firm ; he starred alongside actors Mark Strong and Derek Jacobi . He was cast in the 2005 theatre productions of the Philip Ridley play Mercury Fur , which was performed at the Drum Theatre in Plymouth and the Menier Chocolate Factory in London . He was directed by John Tiffany and starred alongside Ben Whishaw , Shane Zaza , Harry Kent , Fraser Ayres , Sophie Stanton and Dominic Hall . In 2006 , Boulter starred alongside Whishaw in the play Citizenship written by Mark Ravenhill . He appeared on a 2006 episode of the television series , Doctors , followed by a role in the 2007 theatre production of How to Curse directed by Josie Rourke . How to Curse was performed at Bush Theatre in the London Borough of Hammersmith and Fulham . Boulter starred in two films in 2008 , Daylight Robbery by filmmaker Paris Leonti , and Donkey Punch directed by Olly Blackburn . In May 2008 , Boulter made a guest appearance on a two @-@ part episode arc of the television series Waking the Dead , followed by an appearance on the television series Survivors in November 2008 . He had a recurring role in ten episodes of the television series Casualty in 2010 , as " Kieron Fletcher " . Boulter starred in the 2011 film Mercenaries directed by Paris Leonti . = = Career = = = = = 2000 – 2005 = = = In 2000 Boulter had a guest @-@ starring role on the television series The Bill ; he portrayed " Scott Parry " in the episode , " In Safe Hands " . Boulter starred as " Scott " in the play Herons written by Simon Stephens , which was performed in 2001 at the Royal Court Theatre . A review of Boulter 's performance in The Independent on Sunday described him as " horribly menacing " in the role , and he received critical reviews in The Herald , and Evening Standard . He appeared in the television series Judge John Deed in 2002 as " Addem Armitage " in the episode " Political Expediency " , and had a role as a different character " Toby Steele " on The Bill . He had a recurring role in 2003 on two episodes of The Bill , as character " Connor Price " . In 2004 Boulter landed a role as " Craig " in the episode " Teddy 's Story " of the television series The Long Firm ; he starred alongside actors Mark Strong and Derek Jacobi . Boulter starred as " Darren " , in the 2005 theatre productions of the Philip Ridley play Mercury Fur . It was performed at the Drum Theatre in Plymouth , and the Menier Chocolate Factory in London . He was directed by John Tiffany and starred alongside Ben Whishaw , Shane Zaza , Harry Kent , Fraser Ayres , Sophie Stanton and Dominic Hall . Boulter received a favorable review in The Daily Telegraph : " The acting is shatteringly intense , with wired performances from Ben Whishaw ( now unrecognisable from his performance as Trevor Nunn 's Hamlet ) , Robert Boulter , Shane Zaza and Fraser Ayres . " The Guardian noted , " Ben Whishaw and Robert Boulter offer tenderness amid the savagery . " = = = 2006 – present = = = In 2006 Boulter starred in the play Citizenship written by Mark Ravenhill . The play was part of a series which featured different playwrights , titled Burn / Chatroom / Citizenship . In a 2006 interview , fellow actor Ben Whishaw identified Boulter as one of his favorite co @-@ stars : " I loved working with a guy called Robert Boulter , who was in the triple bill of Burn , Chatroom and Citizenship at the National . He played my brother in Mercury Fur . " He portrayed " Jason Tyler " on the 2006 episode of the television series , Doctors , titled " Something I Ate " . Boulter starred as " William " in the 2007 production of How to Curse directed by Josie Rourke . How to Curse was performed at Bush Theatre in the London Borough of Hammersmith and Fulham . In a review of the production for The Daily Telegraph , theatre critic Charles Spencer noted , " Robert Boulter brings a touching vulnerability to the stage as William . " Boulter starred in two films in 2008 , Daylight Robbery by filmmaker Paris Leonti , and Donkey Punch directed by Olly Blackburn . Boulter portrayed a character named " Sean " in Donkey Punch , who tags along with character " Josh " as the " quiet brother ... who hits it off with Tammi " . Boulter guest starred on a two @-@ part episode arc " Wounds " in May 2008 of the television series Waking the Dead as character " Jimmy Dearden " . He appeared on the television series Survivors as " Neil " in November 2008 . He had a recurring role in ten episodes of the television series Casualty in 2010 , as " Kieron Fletcher " . He portrayed an emergency physician applyi

---

## G. Exploring the Trade-offs Between Throughput and Perplexity

In this section, we investigated the relationship between throughput improvements and perplexity across varying sparsity levels. Figure 14 demonstrates how each point corresponds to a specific sparsity level. A sharp increase in perplexity is observed at the 50% sparsity point, highlighting the critical threshold for balancing sparsity and preserving model performance while accounting for the speed tradeoff. Based on this finding, we recommend using OrthoRank with sparsity levels below 40% to maintain an optimal balance.

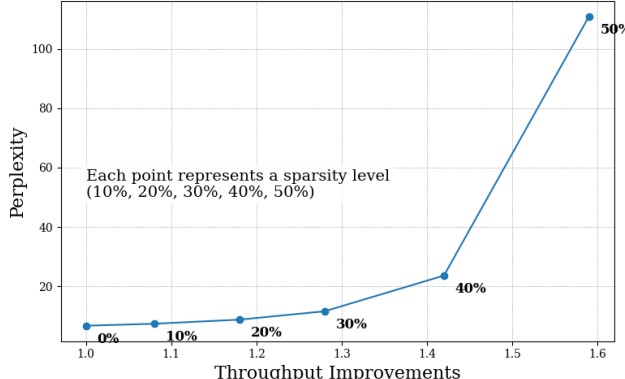

*Figure 14.* The relationship between throughput improvements and perplexity (C4). Each point represents a different sparsity level, showing a sharp increase in perplexity at the 50% sparsity level

# H. Dynamic Token Selection Ratio

This section presents a set of experiments analyzing the effect of varying the token selection ratio across layers while maintaining a fixed average sparsity. The goal is to evaluate the impact of simple heuristics for distributing sparsity without increasing overall budget.

**Experimental Setup**

Experiments were conducted with Llama-2-13B at 20% overall sparsity. Token selection was applied to the following layers:

$$[8, 9, 10, 11, 12, 22, 25, 27, 29, 31, 33, 34]$$

Three strategies for assigning per-layer token selection ratios were evaluated:

- **Fixed**: Constant ratio of 0.33 across all selected layers.

- **Linear1**: Ratio increases linearly with layer depth.
  `[0.0, 0.061, 0.121, 0.182, 0.242, 0.303, 0.364, 0.424, 0.485, 0.545, 0.606, 0.667]`

- **Linear2**: Ratio decreases linearly with layer depth.
  `[0.667, 0.606, 0.545, 0.485, 0.424, 0.364, 0.303, 0.242, 0.182, 0.121, 0.061, 0.0]`

**Results**

Table 9. Performance of different token ratio assignment strategies.

| Method | ppl$_\downarrow$ | acc$_\uparrow$ |
|---|---|---|
| Fixed (uniform ratio 0.33) | **8.74** | 66.99 |
| Linear1 (ratio increases with depth) | 9.10 | 63.74 |
| Linear2 (ratio decreases with depth) | 8.97 | **68.18** |

The fixed-ratio configuration produced the lowest perplexity, while allocating higher token ratios to earlier layers (Linear2) yielded the highest accuracy. These configurations serve as simple baselines for dynamic ratio selection under constrained sparsity.

# I. Example of Generation Task

*Table 10.* Example of generation task.

| Generation using OrthoRank | |
|---|---|
| Prompt for OrthoRank | Struggling with a recent break-up, a person opens up about the intense feelings of loneliness and sadness. They ask for advice on how to cope with the heartbreak and move forward in life. They also wonder if they will ever find love again. |
| OrthorRank | I'm sorry to hear that you are struggling with loneliness after a recent break-up. It's natural to feel sad and lonely after a relationship ends, and it's not uncommon to feel like you are the only one experiencing these feelings. 
 Here are some tips to help you cope with loneliness and move forward: 
 1. Acknowledge your feelings: It's okay to feel lonely and sad. Allow yourself to process your emotions and give yourself time to heal. 
 2. Connect with friends and family: Reach out to loved ones for emotional support and companionship. 
 3. Engage in activities: Try new hobbies or activities that bring joy and help you feel productive, such as exercising, reading, or cooking. 
 4. Practice self-care: Take care of your physical and emotional needs. Get enough sleep, eat well, exercise regularly, and take breaks from screens. 
 5. Seek professional help: If you find that your feelings of loneliness and sadness are persistent and affecting your daily life, consider seeking counseling or therapy. 
 Remember that healing takes time, and it's important to be patient and compassionate towards yourself. With time and effort, you can move forward in life feeling stronger and happier. |
| Evaluation using ChatGPT 4o | |
| Prompt for ChatGPT 4o | Given the following response to the prompt, evaluate its quality on a scale from 1 to 5, considering factors such as relevance, clarity, and accuracy. |
| ChatGPT 4o | I would rate this response **4** out of 5 based on the following factors: 

 **Strengths:** 

 Relevance: The response directly addresses the emotional struggle of a breakup, providing practical and empathetic advice tailored to the prompt. 
 Clarity: The suggestions are well-organized, easy to follow, and actionable, making them accessible for the reader. 
 Empathy: The tone is kind and understanding, which is critical in responding to someone dealing with heartbreak. Accuracy: The advice, including acknowledging feelings, connecting with loved ones, and seeking professional help, aligns with well-established coping strategies. 

 **Areas for Improvement**: 

 Depth: While the advice is solid, it could go further in addressing the specific question of whether they will find love again, which is an integral part of the prompt. 
 Personalization: The response is somewhat generic and could feel more tailored by acknowledging the unique experience of each individual (e.g., "Every breakup is different, but..."). 
 Emotional Validation: While it acknowledges loneliness and sadness, it could do more to validate the fear and uncertainty about the future, such as addressing the difficulty of imagining a new relationship during heartbreak. 

 **Overall**: 

 The response is thoughtful and helpful but could improve by addressing the existential question about finding love again and offering more personalized reassurance. |

