# OpenReview forum: "OrthoRank: Token Selection via Sink Token Orthogonality for Efficient LLM inference"
_ICML.cc/2025/Conference — ICML 2025 poster_

### Official Review · Reviewer_SyT3 · 2025-03-11

**Overall Recommendation:** 3

**Summary:**

This paper introduces OrthoRank, a dynamic token selection method that exploits the relationship between sink tokens and other tokens to improve LLM inference efficiency. The authors observe that as layers deepen in LLMs, the cosine similarity between normalized hidden states of the sink token and other tokens increases, while the sink token's state remains relatively static. Based on this, they propose selecting tokens with greater orthogonality to the sink token for computation, bypassing others except for KV calculations. Experiments demonstrate that OrthoRank achieves lower perplexity and higher zero-shot accuracy compared to baselines.

## update after rebuttal

The authors have addressed my main questions and concerns. For me, this is a self-consistent and complete work. I have also carefully read and am aware of the other reviewers' issues. Overall, I maintain my original borderline accept score.

**Claims And Evidence:**

yes

**Essential References Not Discussed:**

None

**Experimental Designs Or Analyses:**

yes

**Methods And Evaluation Criteria:**

yes

**Other Comments Or Suggestions:**

- see the questions below

**Other Strengths And Weaknesses:**

**pros:**

- Method is applicable across different model architectures and sizes.
- Empirical evidence shows performance improvements over existing layer pruning techniques.
- The paper is well written and easy to follow.

**cons:**

- see the questions below

**Questions For Authors:**

- Why should orthogonality be the primary criterion for token importance rather than other metrics like attention scores, gradient-based importance, or semantic significance...?
- How does the performance vary with different definitions of the sink token? Would using a different token (not the first position) as the reference point yield different or better results?
- The fixed token selection ratio (33%) seems arbitrary. Why not implement an adaptive threshold based on the orthogonality distribution within each layer?
- For more real tasks, how does OrthoRank impact generation quality metrics beyond perplexity, such as coherence, factuality, or hallucination rates?
- How does OrthoRank behave with extremely long context lengths where the sink token's influence might be significantly different?

**Relation To Broader Scientific Literature:**

- This paper extends the concept of attention sinks.
- This paper provides an alternative to layer pruning methods.

**Theoretical Claims:**

The paper's main theoretical claim is the derivation of token importance based on orthogonality to the sink token. It appears sound, assuming the stated approximation that normalized hidden states have approximately equal norm.

---

> ### Author Rebuttal · Authors · 2025-03-29
>
> Thank you very much for your thoughtful and constructive review. Below we provide responses regarding the specific questions you raised. We hope this analysis addresses your concerns and welcome any further feedback.
>
> ### **Q1: Orthogonality as a Token Importance Metric**
> The rationale behind using orthogonality as a criterion is discussed in detail in Q1 from the first reviewer, gDdH. Please refer to it for further clarification.
>
> From an implementation perspective, alternative metrics such as attention scores are less practical. Computing them for unselected tokens introduces overhead, defeating the purpose of selective computation. Moreover, flash attention does not expose intermediate values, further limiting feasibility. Gradient-based methods are also expensive, and semantic significance lacks a clear definition, requiring further exploration.
>
> ### **Q2: Sink Token Definition**
> We experimented with the Llama-2-7B model, as [0] notes that attention sink can also occur at the first strong delimiter (e.g., “.” or “\n”), allowing us to examine the impact of different reference points.
>
>  | Reference| Llama-2-7B (ppl / acc)|
> |-|-|
> | **first position**| **10.04** / **60.35**     |
> | first ".” or “\n” | 10.61 / 59.66  |
>
> Alternative reference points showed similar or slightly lower performance. This is expected, as the cosine similarity between sink tokens at different reference points was very high (over 0.95), resulting in similar selected tokens. The slightly lower performance is likely due to errors caused by unparsed strong delimiters. Since the conditions for sink token vary by model, we recommend using the first position as a reference point.
>
> ### **Q3: Dynamic Token Selection Ratio**
> We believe that your idea represents a crucial and necessary approach. However, optimizing layer selection with varying token selection ratios would greatly expand the search space and require further research. Therefore, we tested a simpler approach by maintaining an average sparsity equal to the fixed 33% ratio through a linear variation in the token selection ratio according to layer depth.
>
> ### **Experimental Setup**
>
> We conducted experiments using the Llama-2-13B model with a target sparsity of 20%. The selected layers for token selection were as follows:
>
> Selected layers for 20% sparsity: [8, 9, 10, 11, 12, 22, 25, 27, 29, 31, 33, 34]
>
> We evaluated two different strategies for the **token selection ratio per layer**:
>
> **Linear1: Selecting more tokens in deeper layers**
>
>    - Token selection ratio:  [0.0, 0.061, 0.121, 0.182, 0.242, 0.303, 0.364, 0.424, 0.485, 0.545, 0.606, 0.667]
>
> **Linear2: Selecting more tokens in shallower layers**
>
>    - Token selection ratio:  [0.667, 0.606, 0.545, 0.485, 0.424, 0.364, 0.303, 0.242, 0.182, 0.121, 0.061, 0.0]
>
> ### **Results**
>
>  | Method| ppl / acc|
> |-|-|
> | **Fixed ratio (0.33)**| **8.74** / 66.99     |
> | Linear 1 | 9.10 / 63.74             |
> | Linear 2 | 8.97 / **68.18**             |
>
> Experimental results showed that the fixed ratio setting achieved better perplexity, while selecting more tokens in shallower layers led to higher accuracy. This demonstrates that even with a simple approach, performance improvements are possible. We believe that determining a dynamic token selection ratio, building on these results, would be a promising research direction.
>
> ### **Q4: Impact on Generation Quality Metrics**
> To address concerns about generation quality, we conducted additional experiments using the TruthfulQA benchmark, which evaluates truthfulness and the ability to avoid hallucinations.
>
> We evaluated two different metrics:
>
> MC1: Measures accuracy by selecting the highest log-probability answer among choices.
>
> Generation (BLEU): Measures truthfulness by comparing the generated response with the ground truth, while also considering informativeness to avoid non-informative answers.
>
> | Method | Llama-2-13B (mc1/ gen) | Llama-3-8B (mc1/ gen) | Mistral-7B (mc1/ gen) | Mixtral-8X7B (mc1/  gen) |
> |-|-|-|-|-|
> | SLEB  | 21.2 /  20.82     | 19.8 / 4.17     | 21.3 / **21.6**    | 24.1 /**27.1** |
> | +OrthoRank | **22.3** / **23.6**             | **21.6** / **15.28**       |  **23.6** / 20.17            | **25.2**/ 26.85       |
>
> The results demonstrate that OrthoRank improves factual accuracy compared to the baseline SLEB model in most cases.
> Generation examples can also be found in Appendix H.
>
> ### **Q5: Performance with Extremely Long Contexts**
> Figure 3, Table 8 in [1] shows that StreamingLLM, which caches initial tokens, consistently outperforms other methods even with long contexts, highlighting the significant influence of the initial sink token. Similarly, Section 4.5 demonstrates that OrthoRank achieves strong performance, indirectly suggesting that the sink token’s influence remains stable even with longer context lengths.
>
> ---
> [0] Massive activations in large language models, COLM 2024
>
> [1] Efficient Streaming Language Models with Attention Sinks, ICLR 2024

---

> > ### Comment · Reviewer_SyT3 · 2025-04-08
> >
> > Thank you for the response. Most of my concerns have been addressed. For now, I will keep my score, and I will also pay attention to the authors' discussion with other reviewers.

---

> > > ### Author Response · Authors · 2025-04-08
> > >
> > > Thank you for your thoughtful response. I truly appreciated your detailed feedback—on the token importance criterion, the definition of sink tokens, the adaptive method, generation quality, and the handling of extremely long contexts—which helped us highlight OrthoRank’s strengths from multiple perspectives and improve the overall quality of the paper.
> > >
> > > We are also pleased to note that **most of your concerns have been addressed**. To provide a **more complete response to Q1**, we include **empirical comparisons** against **attention-based selection** to further support our use of orthogonality beyond its theoretical motivation. The results are shown in the following table.
> > >
> > > | Method |Sparsity | Throughput imporv. | Llama-2-13B (ppl / acc) | Llama-3-8B (ppl / acc) | Mistral-7B (ppl / acc) | Mixtral-8X7B (ppl / acc) |
> > > |-------|-|-|-------------------------|-------------------------|-------------------------|--------------------------|
> > > | **Orthogonal ↑**  |20% |  **1.18x**| **8.74** / **66.99**     | 14.95 / **60.84**    | **11.54** / **63.88**    | 9.39 / **72.52** |
> > > | Attention ↑ | 20% | 0.71x|  8.90 /     63.33           |     **14.85** /  57.76 |  11.60 /      56.81              | **9.01**/  66.73 |
> > >
> > > - To enable the attention-based token selection baseline, we had to use **eager attention** specifically for that method.
> > > - While attention-based selection sometimes shows lower perplexity depending on the model, it consistently results in **significant drops in zero-shot accuracy**.
> > >
> > > - Moreover, we emphasize that **computing attention scores** for all tokens, including **those not selected**, introduces **overhead** and makes the method **incompatible** with **fused-kernel implementations** such as FlashAttention or SDPA, thereby undermining the efficiency gains expected from selective computation.
> > > ---
> > >
> > > Thank you once again for your time, effort, and valuable contributions.
> > >
> > >
> > > Best regards,
> > > Authors

---

### Official Review · Reviewer_Z2Yg · 2025-03-14

**Overall Recommendation:** 3

**Summary:**

This paper joins the rank of other works that are concerned reducing LLM inference costs. The authors start with the observation that the cosine similarity between the hidden states of the sink token and other tokens increases, the deeper in the model we are, despite stationary sink hidden states. Based on that observation, the authors propose OrthoRank which prioritizes the computations of tokens whose hidden states are roughly orthogonal to that of the sink tokens.

**Claims And Evidence:**

The authors make two empirical claims: first that the hidden states of non-sink tokens converge in cosine similarity to the sink token, and second that their approach OrthoRank, derived from latter observation, results in lower perplexity, higher accuracy with comparable throughput. I find the experiments to support all of the above claims.

**Essential References Not Discussed:**

None that I'm aware of

**Experimental Designs Or Analyses:**

I checked all of the experimental details

**Methods And Evaluation Criteria:**

yes

**Other Comments Or Suggestions:**

- (nitpicking) I would suggest modifying Figure 2 as it is currently illegible on paper due to the very small font size

**Other Strengths And Weaknesses:**

- I think the paper is well written and was easy to follow. Moreover, I think the idea is intuitive and I expect it to see a fair share of adoption. That being said, I was left wondering about some experimental details that I would like to ask about:

1. Starting line 237, what model is being used here? and I'm guessing this was performed at every layer?
2. Could you give more details about how you're using OrthorRank with selective layers? How exactly are the layers being selected?
3. In section 4.2, am I to understand that Wikitext-2 was used as a sort of validation dataset? Does your approach require access to a validation set during inference?
4. I was expecting to see comparisons against other methods, including methods that perform token pruning but also approximate attention approaches which, while different in their approach, aim to achieve the same goal as layer pruning and token pruning approaches.

**Questions For Authors:**

Please see above

**Relation To Broader Scientific Literature:**

I think this paper will have an impact in decreasing the inference cost of LLMs as it propose an off-the-shelf approach to saving computation by ignoring tokens that might not need to be updated.

**Theoretical Claims:**

not applicable

---

> ### Author Rebuttal · Authors · 2025-03-28
>
> Thank you for your supportive review and thoughtful suggestions. We greatly appreciate your positive feedback on the clarity, intuition, and practical impact of OrthoRank.
>
> Below we provide responses regarding the specific questions you raised. We hope this analysis addresses your concerns and welcome any further feedback.
>
> ### **Q1: Experimental Details (Line 237)**
> Thank you for pointing it out. The results in line 237 are based on the Llama2-13b model. Token selection was performed at a single layer, and the change in the final output was measured. This process was repeated for each layer to evaluate our criterion. Additional results from other models can be found in Appendix B. We will update the revised manuscript accordingly. Thank you again for your careful review.
>
> ### **Q2: Selective Layer Usage Clarification &  Q3: Use of Wikitext-2 Dataset (Section 4.2)**
> Similar to selecting layers for pruning in layer pruning methods, we progressively replace layers with OrthoRank-applied layers, identifying the layers that exhibit minimal performance degradation (as described in Section 3.2). During this process, we can also utilize an iterative approach, such as SLEB [0], which is represented as "SLEB + OrthoRank" in the experimental results. To assess the impact of applying OrthoRank, a validation set is required, and we used Wikitext for this purpose. Since the layer selection process is conducted offline, there is **no need to access the validation set during actual inference**. In accordance with the evaluation protocol [0], we ensured that performance comparisons were conducted on C4 instead of Wikitext to prevent information leakage from the layer selection process.
>
> ### **Q4: Broader Method Comparisons**
> Approximate attention methods focus on selecting a subset of keys and values to approximate the output of full attention computation. In contrast, as outlined at the end of **Section 3.2**, our proposed OrthoRank reduces **the number of queries**, thereby not only lowering attention computation but also **reducing the input size for the feed-forward network (FFN)**, effectively decreasing the overall computational cost (token selection ratio). Thus, OrthoRank and approximate attention methods can be used together rather than being direct alternatives. We will clarify this distinction in the revised version of our paper.
>
> ### **Figure 2 Readability**
>    Thank you for your note on readability issues with Figure 2. We will improve its visibility by adjusting font sizes and layouts in our updated manuscript.
>
> ---
> [0] SLEB: Streamlining LLMs through Redundancy Verification and Elimination of Transformer Blocks, ICML 2024

---

### Official Review · Reviewer_NuSv · 2025-03-16

**Overall Recommendation:** 2

**Summary:**

The paper introduces a novel dynamic token selection method, OrthoRank, aimed at improving the efficiency of large language model (LLM) inference with fewer computation especially for long context. The authors observe that for some models as layers deepen, the cosine similarity between the normalized hidden states of the sink token and other tokens increases, while the sink token's normalized hidden states remain largely unchanged. Based on this, OrthoRank selects tokens that are more orthogonal to the sink token and assigned greater importance. The authors show that for a previous method STEB or Shortened LLaMA, OrthoRank can improve its performance on longbench.

**Claims And Evidence:**

(1) The nature of sink token: well supported by various size models;
(2) Connecting sink token to token selection: somehow blur, why the orthogonality can be generally used for token selection? It is true that sink tokens tend to be static and finding different representation can bring diversity. However, tokens orthogonal to sinks may also contain excessive noise that should be learned to be eliminated. Are there any constraints needed instead of pursuing pure orthogonality?

**Essential References Not Discussed:**

This paper generally cites references well.

**Experimental Designs Or Analyses:**

The evaluation as well as visualization looks sufficient to characterize the method well.

**Methods And Evaluation Criteria:**

Generally the datasets involved are sufficient to evaluate the methods. However, the comparison with existing layer pruning methods only considers SLEB and Shortened LLaMA. There are way more methods such as H2O, SnapKV, etc. The comparison can cover more methods.

**Other Comments Or Suggestions:**

NA

**Other Strengths And Weaknesses:**

NA

**Questions For Authors:**

Please see the above sections.

**Relation To Broader Scientific Literature:**

The reviewer does not find significant broader scientific contribution.

**Theoretical Claims:**

The paper includes a theoretical derivation to support the token selection criterion based on cosine similarity, which is mainly based on cosine similarity of sink token and other tokens. The intuition makes sense, while the question is about the connection to token selection.

---

> ### Author Rebuttal · Authors · 2025-03-28
>
> Thank you very much for your insightful comments and constructive criticism. We appreciate your careful consideration of our work and your valuable suggestions for improvement.
>
> Below we address the concerns you raised and outline how we will incorporate your feedback:
>
> ### **Connecting sink token to token selection**
>
> We agree that orthogonality can promote diversity and acknowledge the potential for noise. However, our approach is based on the observation that, as tokens propagate through the layers of an LLM, they naturally become more **aligned with the sink token**, increasing their **cosine similarity** (Figure 2, L209-L211). We interpret this growing alignment through layers as an indicator that a token is **on the path of being updated** (Section 3.1). Therefore, we consider tokens with **faster alignment**—i.e., **higher speed**—as more important (L31-L33). Rather than pursuing orthogonality purely for diversity, we use it as a **proxy for speed** and leverage this dynamic for token selection.
>
> In this context, orthogonality is not just for diversity but reflects the natural convergence behavior of tokens. Our experiments show that this approach improves performance without additional training. We appreciate your valuable input and will explore incorporating additional constraints—such as semantic relevance or inter-token relationships—beyond pure orthogonality, to develop a more robust token selection strategy.
>
> Please kindly refer to Q1 with Reviewer gDdH and Q1 with Reviewer SyT3 if you're also interested in comparisons with other criteria.
>
> We will provide a clearer explanation and include the relevant discussion in the revised version of the paper.
>
> ### **Comparison with More Methods**
> H2O and SnapKV are algorithms designed for managing KV caches, which belong to a different research domain compared to OrthoRank. Although our approach differs as we calculate KV even for unselected tokens, it is possible to apply these algorithms to OrthoRank. We believe that by reducing the number of forwarded tokens through OrthoRank and minimizing KV cache using techniques like H2O and SnapKV, the overall efficiency of LLM inference could be significantly improved. We will clearly address the differences between our approach and these studies in the revised manuscript.

---

### Official Review · Reviewer_gDdH · 2025-03-20

**Overall Recommendation:** 3

**Summary:**

The paper introduces OrthoRank, a new method for selecting important tokens in Large Language Models (LLMs) to improve inference efficiency. The method is based on the observation that in LLMs, after the attention sink occurs, the cosine similarity between the normalized hidden states of the sink token and other tokens increases as layers deepen, while the sink token's hidden states remain relatively unchanged.  OrthoRank selects tokens based on their orthogonality to the sink token, prioritizing tokens that are more orthogonal for updates. The authors claim that OrthoRank achieves lower perplexity and higher zero-shot accuracy compared to layer pruning methods at the same sparsity ratio, with comparable throughput, and superior performance on LongBench.

**Claims And Evidence:**

The cosine similarity analysis and the behavior of sink tokens are well-supported. The authors present detailed results in Section 2 and Appendix B, with clear visualizations (Figures 2, 3, 9, and 10) that corroborate their observations regarding the cosine similarity between the sink token and other tokens, and the relatively unchanged state of the sink token across layers.

The effectiveness of OrthoRank is demonstrated through various experiments. The authors compare OrthoRank with layer pruning methods and show that it achieves better performance in terms of perplexity, zero-shot accuracy, and performance on LongBench.  The ablation studies further validate the design choices of OrthoRank, such as the token selection criteria and the importance of KV calculations.

The trade-offs between throughput and performance are analyzed. Figure 6 and Figure 14 illustrate the relationship between throughput improvements and perplexity, showing that OrthoRank achieves comparable or better performance than layer pruning methods while maintaining a throughput increase nearly proportional to sparsity.

While the paper demonstrates OrthoRank's superior performance compared to layer pruning, the discussion around the limitations of layer pruning could be more nuanced. The paper states that layer pruning methods "do not effectively reflect the specific characteristics of the input tokens"  and that they may result in "abrupt performance degradation". While this is true, layer pruning is a well-established and effective technique for LLM efficiency. A more balanced discussion acknowledging the strengths and weaknesses of both approaches would provide a more comprehensive view.

**Essential References Not Discussed:**

Nope

**Experimental Designs Or Analyses:**

The analysis is thorough and well-presented. The authors used appropriate metrics (perplexity, zero-shot accuracy, accuracy on LongBench) to evaluate the performance of OrthoRank. One potential area where the analysis could be enhanced is the discussion around the limitations and trade-offs of OrthoRank. While the authors compare OrthoRank with layer pruning, a more in-depth analysis of scenarios where layer pruning might be more suitable or efficient would provide a more balanced perspective.

**Methods And Evaluation Criteria:**

Proposed Methods:
The proposed method, OrthoRank, is designed to improve the efficiency of Large Language Model (LLM) inference by selecting important tokens and bypassing computations for less important ones.  This is a relevant goal in the context of LLMs, where computational cost is a significant challenge. The method leverages the concept of the "attention sink" and introduces a novel approach to token selection based on the orthogonality of tokens to the sink token. This is a reasonable approach, as it tries to exploit the internal mechanisms of LLMs to achieve efficiency gains.

Evaluation Criteria:
The paper uses perplexity and zero-shot accuracy as key evaluation metrics. These are standard and widely accepted metrics for evaluating language models, making them appropriate for the task. The authors also evaluate their method on the LongBench benchmark, which is designed to assess the performance of models on long-context understanding. This is particularly relevant given the challenges LLMs face with long sequences. The paper includes ablation studies to analyze the impact of different components of their method, such as token selection criteria and the use of KV calculations.  This is a good practice to validate design choices and understand the contribution of each component.

**Other Comments Or Suggestions:**

The authors may want to clarify the positioning of their work with respect to other token selection methods. While they mention that their method does not progressively drop tokens across layers, like some token pruning methods, the distinction could be further emphasized. For example, OrthoRank could be characterized as a method that performs token selection within a layer, maintaining the full sequence length across layers, but reducing computation at selected layers.

**Other Strengths And Weaknesses:**

Strengths:

Novelty: Introduces OrthoRank, a new token selection method based on token-sink orthogonality, and provides new insights into the attention sink phenomenon.

Significance: Addresses the critical problem of high computational cost in LLM inference, contributing to the development of more efficient LLMs.

Clarity: Well-written and easy to follow, with clear explanations, figures, and supplementary materials.

Weaknesses:

Scope of Analysis: Primarily focuses on the relationship between the sink token and other tokens, with limited analysis of relationships among non-sink tokens.

Assumption of Equal Norms: Relies on the assumption that normalized hidden states have approximately equal norms, which could benefit from further theoretical justification.

Generalizability: Primarily demonstrates effectiveness on autoregressive models; generalizability to other LLM types could be explored further.

**Questions For Authors:**

Token Selection Rationale: Can you provide more insight into why orthogonality-based token selection is more effective than alternatives like attention scores or hidden state magnitudes?

Layer Pruning Limitations: Could you elaborate on the trade-offs between OrthoRank and layer pruning, discussing when layer pruning might be preferred?

Generalizability: How can OrthoRank be applied to LLMs beyond autoregressive models, such as encoder-decoder models?

Hyperparameter Tuning: Please provide more guidance on tuning OrthoRank's hyperparameters (e.g., token selection ratio, layer selection) for optimal performance.

**Relation To Broader Scientific Literature:**

Attention Sink Analysis: The paper builds upon existing research on the "attention sink" phenomenon in LLMs.
It references the initial discovery of the attention sink by Xiao et al. (2024), which highlighted how the initial token in a sequence often receives disproportionately high attention. It also acknowledges further explorations of this phenomenon and techniques to calibrate or leverage it for improved LLM efficiency. The authors expand on this by analyzing the cosine similarity between the sink token and other tokens in hidden states, which they claim is a novel approach.

Token Selection Methods: The paper's contribution to token selection is related to prior work in token pruning and dynamic token selection. It contrasts its approach with token pruning methods that progressively drop tokens across layers.
It also positions its work in the context of dynamic token selection and early exit mechanisms, which aim to improve efficiency by selectively processing tokens. The key difference is that OrthoRank selects tokens based on their orthogonality to the sink token, without requiring additional training or modules.

Efficiency in LLMs: The paper addresses the broader challenge of improving the efficiency of Large Language Models (LLMs), which is a significant area of research. It discusses layer pruning as a common technique for reducing computational costs. It contrasts OrthoRank with layer pruning, highlighting its ability to provide more fine-grained control over computational efficiency by selecting tokens within layers.

**Theoretical Claims:**

The paper includes a section (Section 3.1) that provides a derivation for its token selection criteria. Here's a breakdown of my assessment:

The authors aim to define token importance based on the change in cosine similarity with the sink token. They start by expressing the cosine similarity and computing its gradient with respect to the hidden state of a token. The derivation seems correct.

To simplify the importance metric, the authors make an assumption that normalized hidden states have approximately equal norms (except for the sink token). This assumption is supported by Figure 12 in Appendix C, which shows that the norms of hidden states are indeed approximately equal.

Based on this assumption, the authors simplify the expression and show that the importance of a token is related to how small the cosine similarity between that token and the sink token is. The algebraic manipulations in Appendix C appear to be correct.

Overall, the derivation of the token selection criteria seems to be logically sound and the assumption made is supported by empirical evidence.

---

> ### Author Rebuttal · Authors · 2025-03-28
>
> Thank you very much for your detailed and constructive feedback. We truly appreciate the considerable time and effort you invested in evaluating our paper and your recognition of OrthoRank’s novelty. Below we clarify the points you raised.
>
> ### **W2: Assumption of Equal Norms**
> OrthoRank is based on (scaled) normalized hidden states obtained via RMSNorm, which first normalizes each token vector to unit RMS norm and then applies a learned element-wise scale:
> $$
> \mathbf{h}_i^{\text{scaled}} = \mathbf{g} \odot \left( \frac{\mathbf{h}_i}{\text{RMS}(\mathbf{h}_i)} \right) = \mathbf{g} \odot \mathbf{h}_i^{\text{norm}}
> $$
> This ensures all tokens start with equal norm, after which **learned scaling** introduces norm differences. Interestingly, the scaling vector **g** has **very low variance (~0.001)**, effectively acting as a near-constant scalar. For **non-sink tokens**, cosine similarity between scaled and normalized hidden states remains high (≥ 0.95), indicating that scaling behaves approximately as **scalar multiplication**:
> $$
> \cos(\theta) \approx 1 \quad \Rightarrow \quad \mathbf{h}_i^{\text{scaled}} \approx \alpha \mathbf{h}_i^{\text{norm}}
> $$
> Since scalar multiplication preserves norm ratios, the scaled hidden states of non-sink tokens maintain near-equal norms:
> $$
> \|\mathbf{h}_i^{\text{scaled}}\| \approx \alpha \cdot c \approx \|\mathbf{h}_j^{\text{scaled}}\| \quad \text{for all } i, j \text{ in non-sink tokens}
> $$
> In contrast, sink tokens show a significant drop in similarity (~0.6), likely due to **energy compaction** from **massive activation [0]** in a few dimensions. This makes them more sensitive to suppression, even if **g is nearly constant**.
>
> Therefore, given the **uniformity of g**, it is reasonable to assume scaled hidden states of nom-sinks have approximately equal norms.
>
> ### **W3 & Q3: Generalizability**
> To test generalizability beyond autoregressive models, we evaluated OrthoRank on the encoder-based BERT-base-uncased. Attention sinks were observed, most often at [SEP], but also at [CLS] or '.' depending on the layer. However, unlike in autoregressive models, cosine similarity with [SEP] did not show a consistent increase. Thus, OrthoRank is not directly applicable to encoder models, and adapting it may require handling layer-wise variation in sink tokens—a direction we leave for future work.
>
> ### **Q1: Token Selection Rationale**
> We consider an increase in **cosine similarity** with the sink token as an indication that a **token is on the path of being updated**. (Figure 2, L209-L211) Therefore, as tokens propagate through the layers, it is most efficient to update those that exhibit **faster movement**. Based on this intuition, we define the **speed** of a token as its importance. Accordingly, **orthogonality**—interpreted as speed—is used as a criterion for token selection.
>
> The attention score-based KV cache eviction treats tokens as sources of information, aiming to retain those with high attention to surrounding tokens and preserve the attention matrix before and after eviction. While **high attention scores indicate strong influence on other tokens**, they hold **little meaning** from the **updating token’s perspective**.
>
> We lacked insight into using the hidden state norm for token selection, so we conducted experiments to evaluate its effectiveness.
>
> | Method | Llama-2-13B (ppl / acc) | Llama-3-8B (ppl / acc) | Mistral-7B (ppl / acc) | Mixtral-8X7B (ppl / acc) |
> |-|-|-|-|-|
> | Ours  | **8.74** / **66.99**| **14.95** / **60.84** | **11.54** / 63.88| **9.39** / **72.52** |
> | Norm ↑ | 9.46 / 62.94| 16.23 / 59.46|12.26 / 59.97|9.39 / 70.89|
> | Norm ↓ | 9.15 / 64.47| 16.73 / 60.13|21.12 / **63.99**|9.78 / 66.50|
>
> Our method consistently performs well across models, achieving top or comparable results in both perplexity and accuracy. While Mistral-7B slightly outperforms in zero-shot accuracy using the high-norm method, it suffers from higher perplexity, potentially harming performance elsewhere. This highlights the robustness of our approach.
>
> ### **Q2: Balanced Discussion on Layer Pruning**
> We agree that layer pruning is a well-established and effective approach for improving LLM efficiency. As noted, both OrthoRank and layer pruning have distinct strengths and limitations. Layer pruning is particularly advantageous in scenarios with limited storage or where small batch sizes make weight transmission a bottleneck, such as on-device AI. In contrast, when such constraints are absent, OrthoRank—which loads full weights but processes only a subset of tokens—can be more suitable. Each method, therefore, has its own context-dependent benefits.
>
> ### **Q4: Hyperparameter Tuning**
> We recommend setting the token selection ratio between 0 and 0.5 (Section 4.6.4). After determining the desired total sparsity, the layer selection ratio is automatically calculated. We recommend setting the total sparsity to be below 40% (Appendix G)
>
> ---
> [0] Massive activations in large language models, COLM 2024

---

### Decision · Program_Chairs · 2025-05-01

**Decision:**

Accept (poster)

**Comment:**

This paper introduces OrthoRank, a dynamic token selection method to boost inference efficiency of Large Language Models for long contexts. Experiments show lower perplexity, higher zero-shot accuracy, and better LongBench performance than baselines.

Most reviewers agree that the proposed approach is novel and technically solid, and this paper is well-written. During the reviewer-author discussion phase, most of the reviewers tend to accept this paper.

Overall, this paper is valuable for the community. Therefore, I recommend accepting this paper.